# The non-canonical poly(A) polymerase *FAM46C* acts as an onco-suppressor in multiple myeloma

Seweryn Mroczek [1,2], Justyna Chlebowska[1,3,4,5], Tomasz M. Kuliński[2], Olga Gewartowska[1,2], Jakub Gruchota[2,6], Dominik Cysewski[2], Vladyslava Liudkovska[1], Ewa Borsuk[7], Dominika Nowis[4,5] & Andrzej Dziembowski [1,2]

*FAM46C* is one of the most frequently mutated genes in multiple myeloma. Here, using a combination of in vitro and in vivo approaches, we demonstrate that *FAM46C* encodes an active non-canonical poly(A) polymerase which enhances mRNA stability and gene expression. Reintroduction of active *FAM46C* into multiple myeloma cell lines, but not its catalytically-inactive mutant, leads to broad polyadenylation and stabilization of mRNAs strongly enriched with those encoding endoplasmic reticulum-targeted proteins and induces cell death. Moreover, silencing of *FAM46C* in multiple myeloma cells expressing WT protein enhance cell proliferation. Finally, using a FAM46C-FLAG knock-in mouse strain, we show that the *FAM46C* protein is strongly induced during activation of primary splenocytes and that B lymphocytes isolated from newly generated *FAM46C* KO mice proliferate faster than those isolated from their WT littermates. Concluding, our data clearly indicate that *FAM46C* works as an onco-suppressor, with the specificity for B-lymphocyte lineage from which multiple myeloma originates.

[1] Institute of Genetics and Biotechnology, Faculty of Biology, University of Warsaw, Pawinskiego 5a, 02-106 Warsaw, Poland. [2] Institute of Biochemistry and Biophysics, Polish Academy of Sciences, Pawinskiego 5a, 02-106 Warsaw, Poland. [3] Department of Immunology, Center of Biostructure Research, Medical University of Warsaw, Banacha 1a, 02-097 Warsaw, Poland. [4] Laboratory of Experimental Medicine, Center of New Technologies, University of Warsaw, Banacha 2c, 02-097 Warsaw, Poland. [5] Genomic Medicine, Medical University of Warsaw, Banacha 1a, 02-097 Warsaw, Poland. [6] International Institute of Molecular and Cell Biology, Trojdena 4, 02-109 Warsaw, Poland. [7] Department of Embryology, Institute of Zoology, Faculty of Biology, University of Warsaw, Miecznikowa 1, 02-096 Warsaw, Poland. Correspondence and requests for materials should be addressed to A.D. (email: andrzejd@ibb.waw.pl)

Mass sequencing of cancer genomes has revealed genomic landscapes of human cancers allowing for the identification of a large number of potential tumor suppressors and oncogenes. *FAM46C*, a gene whose physiological function was largely unknown, is one of the most frequently mutated genes in multiple myeloma (MM), following the well-known proto-oncogenes, *KRAS*, *NRAS*, and *BRAF*[1, 2]. Deletions of *FAM46C* gene 1p12 locus (del(1p)) have been found in ~20% of MM cases and are associated with short progression-free survival and decreased overall survival[3, 4]. Except of chromosomal aberrations, recurrent homozygotic or hemizygotic somatic point mutations have been identified in about 10% of MM cases, depending on studies based on whole-genome- or whole-exome sequencing[3, 5–8]. To date, more than 70 unique somatic mutations across whole *FAM46C* gene sequence have been identified, many of which are frameshift or nonsense mutations (https://research.themmrf.org)[1]. Importantly, FAM46C mutations are specific to MM since no other cancer type with statistical significant enriched in FAM46C mutations has been described so far[9]. The high frequency of mutation in the *FAM46C* gene allowed it to be classified as MM driver-gene, which may function as a tumor suppressor even though it does not contain mutational hotspots[6, 10, 11]. *FAM46C* mutations are also frequently found in stable human myeloma cell lines. In addition, *FAM46C* has been identified as a type I interferon-stimulated gene, overexpression of which slightly enhances replication of some viruses[12, 13].

FAM46C belongs to a FAM46 metazoan-specific family of proteins, which has 4 members in humans that are very similar at the protein level with a degree of sequence identity of at least 56.9%. There is currently very little functional data on FAM46 proteins. Positional cloning in mouse revealed that mutations in *FAM46C* gene cause anemia[14]. The only publication about this phenomenon is the PhD thesis of Tian[14]. The author performed initial biochemical characterization of FAM46C and concluded that it is a RNA-binding protein that stabilizes specific mRNAs in reticulocytes, including that of alfa-globin, which correlates with poly(A) tail shortening. Recent bioinformatic fold recognition searches classified FAM46C as a member of the novel nucleotidyltransferases (NTases) family; however, this study did not provide reliable predictions of molecular function[15, 16]. NTases transfer nucleoside monophosphate (NMP) from nucleoside triphosphate (NTP) to an acceptor hydroxyl group and are involved in many biological processes, including mRNA polyadenylation and editing, DNA repair and chromatin remodeling, intracellular signal transduction, and regulation of protein activity[15, 17].

Here we performed the first comprehensive molecular characterization of *FAM46C*. We show that FAM46C is an active poly(A) polymerase that positively regulates expression of ER-targeted mRNAs. Furthermore, the presented data strongly indicate that FAM46C is a B-cell lineage-specific growth suppressor as silencing of *FAM46C* in MM cells that express the wild-type protein enhances cell division, whereas introduction of wild-type FAM46C into MM that express the protein with mutations leads to growth arrest; also, primary B cells isolated from *FAM46C* KO animals proliferate faster. Thus, we describe *FAM46C* as an onco-suppressor non-canonical poly(A) polymerase.

## Results

**FAM46C encodes a poly(A) polymerase that enhances gene expression.** The FAM46 family of proteins exists only in animals. In vertebrates, all its members have the same architecture. They contain domains that are very distantly related to the catalytic and associated domains of poly(A) polymerases, and they lack any detectable RNA-binding domains[15]. The putative catalytic residues of FAM46C are preserved, indicating that this protein may indeed be an active poly(A) or poly(U) polymerase (Supplementary Fig. 1).

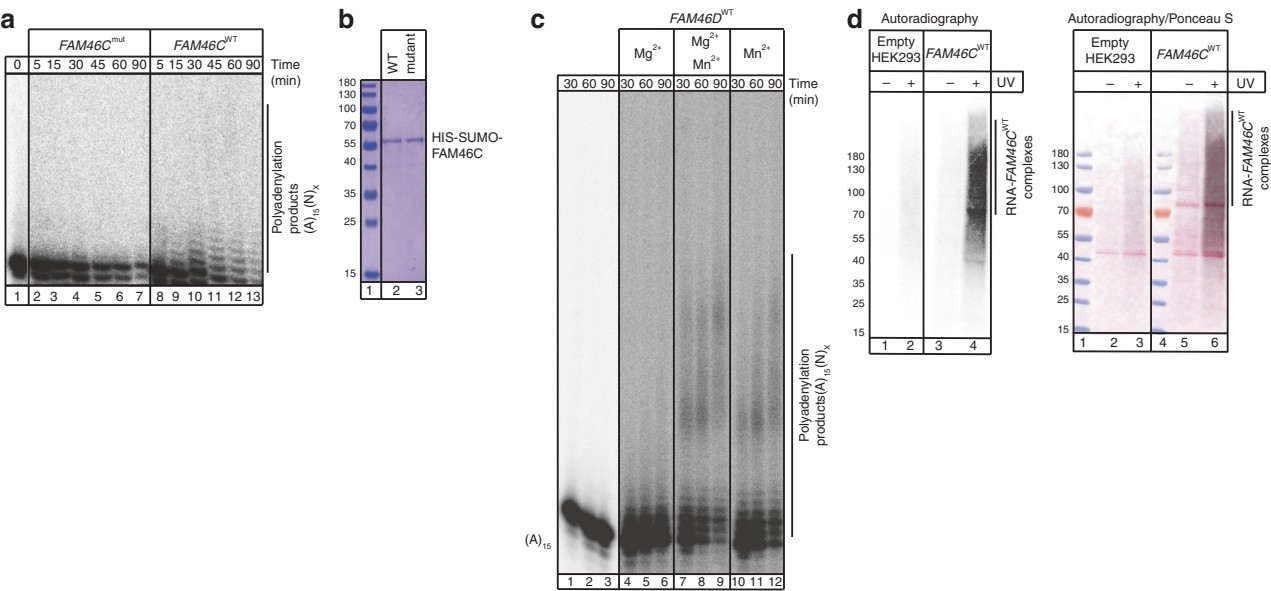

**Fig. 1** FAM46C interacts with RNA and is an active RNA poly(A) polymerase in vitro and in vivo. **a** Recombinant FAM46C[WT] (lanes 8–13), but not its catalytic mutant FAM46C[mut] (lanes 2–7), displays poly(A) polymerase activity in vitro. Reaction products (using [32]P-labeled (A)$_{15}$ as substrate) were separated in denaturing PAGE gels and visualized by autoradiography. **b** SDS-PAGE analysis of recombinant FAM46C[WT] and its catalytic mutant FAM46C[mut]. **c** FAM46D[WT] is an active poly(A) polymerase in vitro and requires Mn$^{2+}$ ions for its activity. Purified protein was incubated with [32]P-labeled (A)$_{15}$ primer in the presence of ATP and divalent cations as follows: Mg$^{2+}$ (lanes 4–6), both Mg$^{2+}$/Mn$^{2+}$ (lanes 7–9), or Mn$^{2+}$ (lanes 10–12). Control reactions were carried out without the protein (lanes 1–3). **d** FAM46C interacts with RNA in human cells. Autoradiography of UV cross-linked [32]P-labeled RNAs co-purified with FAM46C[WT]GFP from HEK293 cells stably expressing the fusion protein (lanes 3–4) and from control empty cells (lanes 1–2). Immunoprecipitated RNA-protein complexes were separated by SDS-PAGE, transferred to nitrocellulose membrane, stained with Ponceau S, and subsequently autoradiographed. The *right* panel shows the Ponceau S stained blot merged with autoradiogram

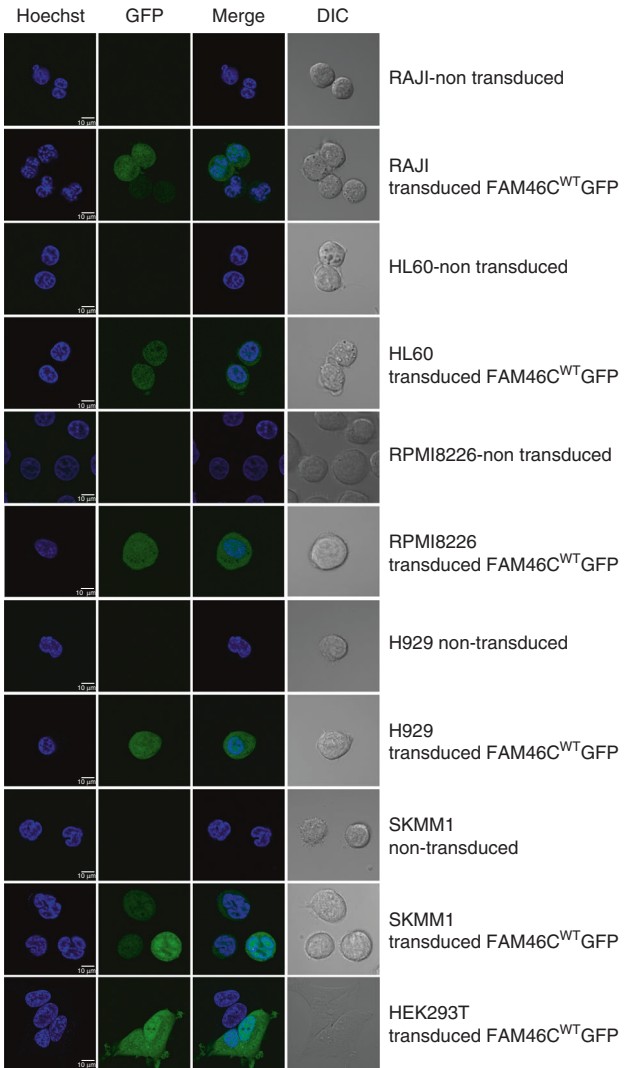

|   | Hoechst | GFP | Merge | DIC |

**Fig. 2** FAM46C shows both nuclear and cytoplasmic localization. HEK293T, RPMI826 (MM), H929 (MM), SKMM1 (MM), HL60 (promyelocytic leukemia) and Raji (B-cell lymphoma) cell lines were transduced with constructs encoding the *FAM46C*^WT–GFP fusion gene. Non-transduced cells were used as a control. Fluorescence images for GFP and Hoechst nucleic acid stain, as well as merged images, are shown. *Scales bars* (10 μm) are shown for each set of images

In order to confirm predicted molecular activity, we purified recombinant FAM46C and its mutant version, in which acidic residues of the putative catalytic center, expected to coordinate divalent cations, were replaced by alanines (D90A and D92A; *FAM46C*^mut). This was followed by an in vitro polyadenylation assay using a $A_{15}$ oligoribonucleotide RNA as a substrate. Wild type, but not catalytic mutant, protein was able to extend the RNA substrate in vitro in the presence of ATP and divalent metal ions ($Mn^{2+}$) (Fig. 1a, b), indicating that FAM46C is a poly(A) polymerase. However, due to the relatively low activity of recombinant FAM46C (likely related to its intrinsic low solubility and aggregation), we validated the result using recombinant FAM46D, which was more soluble; the sequences of the two proteins are highly similar (Supplementary Fig. 1). FAM46D displayed poly(A) polymerase activity and was able to extend $A_{15}$ oligoribonucleotide substrate; its activity was significantly higher compared to that of FAM46C (Fig. 1c).

In order to determine the molecular function of FAM46C, we evaluated whether this protein could interact with RNA at the cellular level. Stable HEK293 cells expressing FAM46C^WT-GFP were UV cross-linked and the tagged proteins were immunoprecipitated using anti-GFP antibodies. Co-purified RNA was visualized by $^{32}$P-labeling of RNA-protein covalent crosslinks. Strong enrichment of RNA was observed in cross-linked, FAM46C-positive samples compared to control cells and non-crosslinked samples (Fig. 1d), demonstrating that FAM46C binds to RNA and may act as an RNA poly(A) polymerase in human cells.

Non-canonical poly(A) polymerases may enhance overall gene expression by extending the poly(A) tails of dormant messenger RNAs (like GLD2) in the cytoplasm or trigger rapid exosome-mediated RNA decay by oligoadenylation of target RNA in the nucleus (as in the case of TRF4)[18]. In order to examine intracellular localization of the FAM46C protein, HEK293T, RPMI826 (MM), H929 (MM), SKMM1 (MM), HL60 (promyelocytic leukemia) and Raji (B-cell lymphoma) cell lines were transduced with constructs encoding the *FAM46C*^WT-GFP fusion gene. Confocal microscopy revealed simultaneous nuclear and cytoplasmic localization of FAM46C in all tested cell lines, as expected for non-canonical poly(A) polymerases (Fig. 2), demonstrating the possibility of both RNA stabilizing and destabilizing functions.

To further analyze FAM46C function in cells, we performed an RNA-tethering assay. HEK293 cells were co-transfected with a construct expressing *Renilla* luciferase (RL) containing five boxB sites in its 3′-UTR, Firefly luciferase (FL) control reporter and *FAM46C*^WT harboring the N-terminal λN boxB-binding domain and HA-tag[19]. After 24 h, the levels of reporter protein were analyzed, which revealed that tethering a wild-type *FAM46C* enhances expression of RL more than 7-fold (Fig. 3a, d). Several additional controls were applied to prove that the catalytic activity of FAM46C was indeed responsible for the enhanced expression of RL reporter: (1) tethering of the catalytic FAM46C mutant had little effect on RL reporter expression (Fig. 3a, d); (2) a RL reporter with a cyclic phosphate at the 3′-end generated by a hammerhead ribozyme, which cannot be polyadenylated, was insensitive to FAM46C tethering (Fig. 3b, d); (3) expression of *FAM46C* without the λN domains did not enhanced expression of the reporter (Fig. 3c, d). At the RNA level, northern blot analysis revealed that enhanced expression of RL upon FAM46C^WT tethering correlated with increased steady state levels and slower migration of its mRNA (Fig. 3e, f). This was not observed for the RL reporter with a cyclic phosphate at the 3′ (Fig. 3e, f). Quantifications of control FL reporter mRNA revealed no effect of FAM46C on its expression (Fig. 3g). Furthermore, northern blot analysis of poly(A) enriched-RNA fractions revealed significant increases in steady-state levels and slower migration of RL mRNA in samples where FAM46C^WT was tethered (Fig. 3h). Next, we confirmed that the observed RL mRNA lengthening is indeed caused by polyadenylation of mRNA 3′-end using an RNase H cleavage assay with oligo(dT)$_{25}$ (Fig. 3i). In addition, RNA from cells after FAM46C was tethered to RL mRNA was fractionated based on the lengths of poly(A) tails using oligo dT sepharose resin. Northern blot analysis revealed that more RL mRNA molecules were present in fractions with longer poly(A) tails, confirming that it is polyadenylated by FAM46C (Supplementary Fig. 2a). The same effect was observed for FAM46D indicating that they have similar properties (Supplementary Fig. 2b).

In order to determine whether the observed increase in reporter expression was due to mRNA stability, we evaluated the half-life of RL mRNA. Cells were transfected with pRL-5BoxB and pNHAFAM46C^WT or pNHAFAM46C^mut. After 24 h, transcription was inhibited for 4, 8, and 12 h by the addition of actinomycin D. Northern blot, followed by densitometry

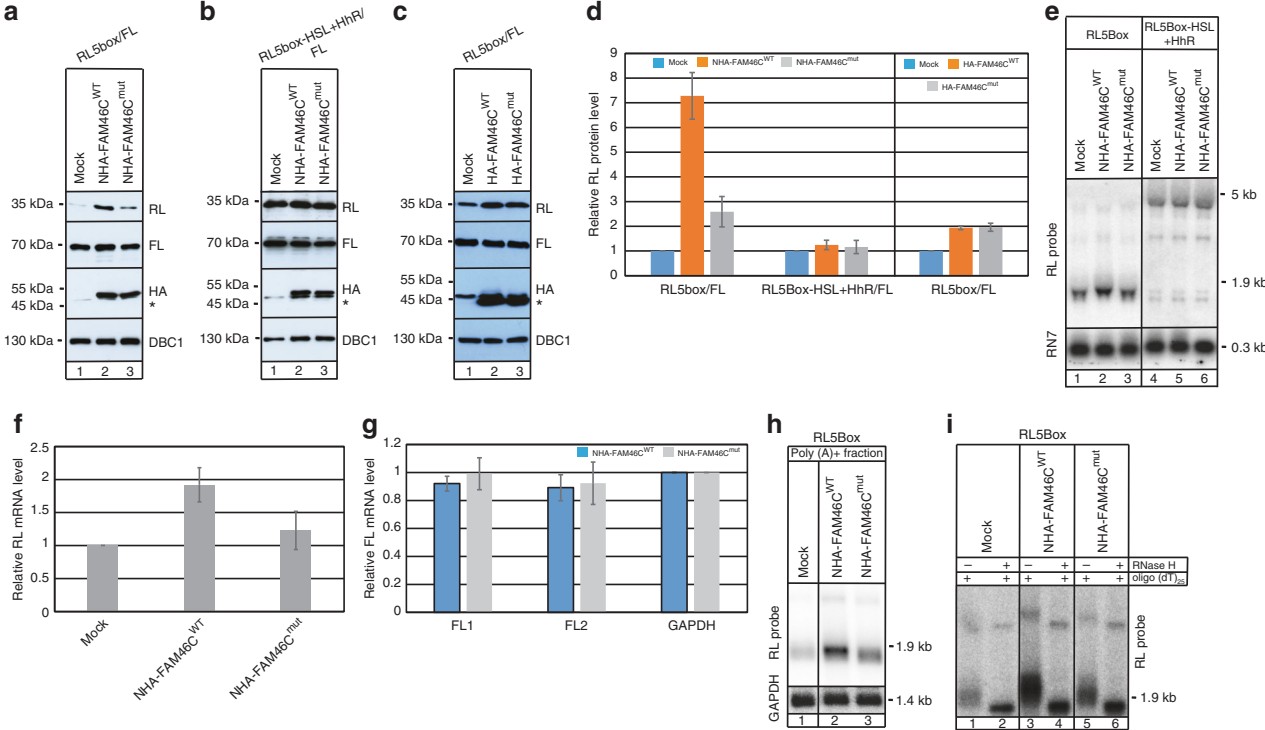

**Fig. 3** FAM46C tethering leads to polyadenylation and enhanced expression of a *Renilla* luciferase (RL) reporter. Analysis at the protein level (**a–d**): **a** FAM46C tethering increases RL reporter protein levels. HEK293 cells were co-transfected with a construct expressing RL, containing five boxB sites in its 3′-UTR, Firefly luciferase (FL) control reporter and FAM46C$^{WT}$ harboring the N-terminal λN boxB-binding domain and HA-tag. Western blot detection of RL in mock-transfected cells (lane 1) and after NHA-FAM46C$^{WT}$ (lane 2) or NHA-FAM46C$^{mut}$ (lane 3) tethering. Expression of NHA-tagged FAM46C proteins were confirmed using an α-HA antibody. DBC1 served as a loading control. *Asterisks* indicate cross-hybridization signals. **b** Expression of RL5boxHSL + HhR reporter with a cyclic phosphate at the 3′ end generated by a hammerhead ribozyme was not enhanced upon FAM46C tethering. The experiment was performed as in **a**. **c** Expression of the FAM46C without λN domains did not enhance expression of the reporter. The experiment was performed as in **a**. **d** Quantifications of RL protein normalized by the internal FL reporter related to experiments from **a–c**. RNA analysis (**e–i**): **e** northern blot detection of *RL mRNA* using total RNA from HEK293 cells after tethering of NHA-FAM46C$^{WT}$ or NHA-FAM46C$^{mut}$ to RL5box (lanes 1–3) or RL5boxHSL + HhR with a cyclic phosphate at the 3′ end (lanes 4–6). **f** Quantifications of RL mRNA. **g** RT-qPCR analysis of the FL control reporter. **h** Northern blot detection of *Renilla* luciferase using poly(A) + fraction from HEK293 cells after tethering of NHA-FAM46C$^{WT}$ or NHA-FAM46C$^{mut}$ (**i**) Poly (A) tails added to reporter mRNA can be removed by RNase H treatment in the presence of oligo(dT)$_{25}$. High-resolution northern blot analysis of RL mRNA from control HEK293 cells (lanes 1–2), after tethering of NHA-FAM46C$^{WT}$ (lanes 3–4) or NHA-FAM46C$^{mut}$ (lanes 5–6). The data in **d**, **f**, **g** are shown as a mean value ± SD ($n = 3$)

quantifications analyses of steady state levels of RL mRNA obtained from appropriate time points, revealed significant increases in the mRNA half-life of the molecules polyadenylated by FAM46C compared to control cells (Supplementary Fig. 2c, d). Therefore, it was concluded that FAM46C is an active poly(A) polymerase which, when tethered, stabilizes mRNA and enhances gene expression.

**FAM46C is an MM cell growth suppressor.** To study *FAM46C* in the context of MM pathogenesis, we took advantage of the established MM cell lines, some of which harbor early frameshift mutations in *FAM46C*: SKMM1 (Homozygous p.I173fsX36) and H929 (hemizygous p.L93fsX15) (Supplementary Table 1). In order to reintroduce wild-type *FAM46C* into these cell lines, we designed a lentiviral vector allowing for the expression of C-terminally GFP-tagged protein under the control of the spleen focus-forming virus (SFFV) promoter, which is active in hematopoietic cells[20]. Initially, we evaluated transduction efficiency and protein expression levels using control HEK293T cells, which were transduced with lentiviruses carrying *FAM46C*$^{WT}$-GFP, *FAM46C*$^{mut}$-GFP, or GFP as control, using different multiplicities of infection (MOIs; from 1 to 4). In all cases, stable cell

lines were obtained with >95% efficiency as verified by flow cytometry for GFP-expressing living cells (Supplementary Fig. 3a). No significant growth phenotypes were observed in HEK293T cells. A cytometric analysis of the cell cycle revealed higher number of cells in the G0/G1 phase; however, this observation could be a side effect associated with lentiviral particles, as it was not correlated with the activity of the FAM46C protein.

Next, we transduced SKMM1, H929 (both mutated in the endogenous *FAM46C* locus), and RPMI8226 (with wild-type *FAM46C* sequence) with lentiviruses carrying *FAM46C*$^{WT}$-GFP, *FAM46C*$^{mut}$-GFP, and GFP constructs at MOIs from 1 to 4. Cytometric analysis of cell cultures was used to monitor the effect of FAM46C expression up to 11 days post transduction. Three days after transgene delivery, most of the tested cell lines expressing *FAM46C*$^{WT}$ had reduced growth rate and increased number of dead cells (Fig. 4a, b); this effect was positively correlated with high FAM46C$^{WT}$ virus titer levels (Supplementary Fig. 4b). Interestingly, in SKMM1 cells, the cytostatic effect of *FAM46C*$^{WT}$ expression was observed later in comparison to H929 or RPMI8226 cells and was best visible from the ninth day after transduction. Cytostatic effect was also observed for constructs with other tags at the N and C termini of FAM46C

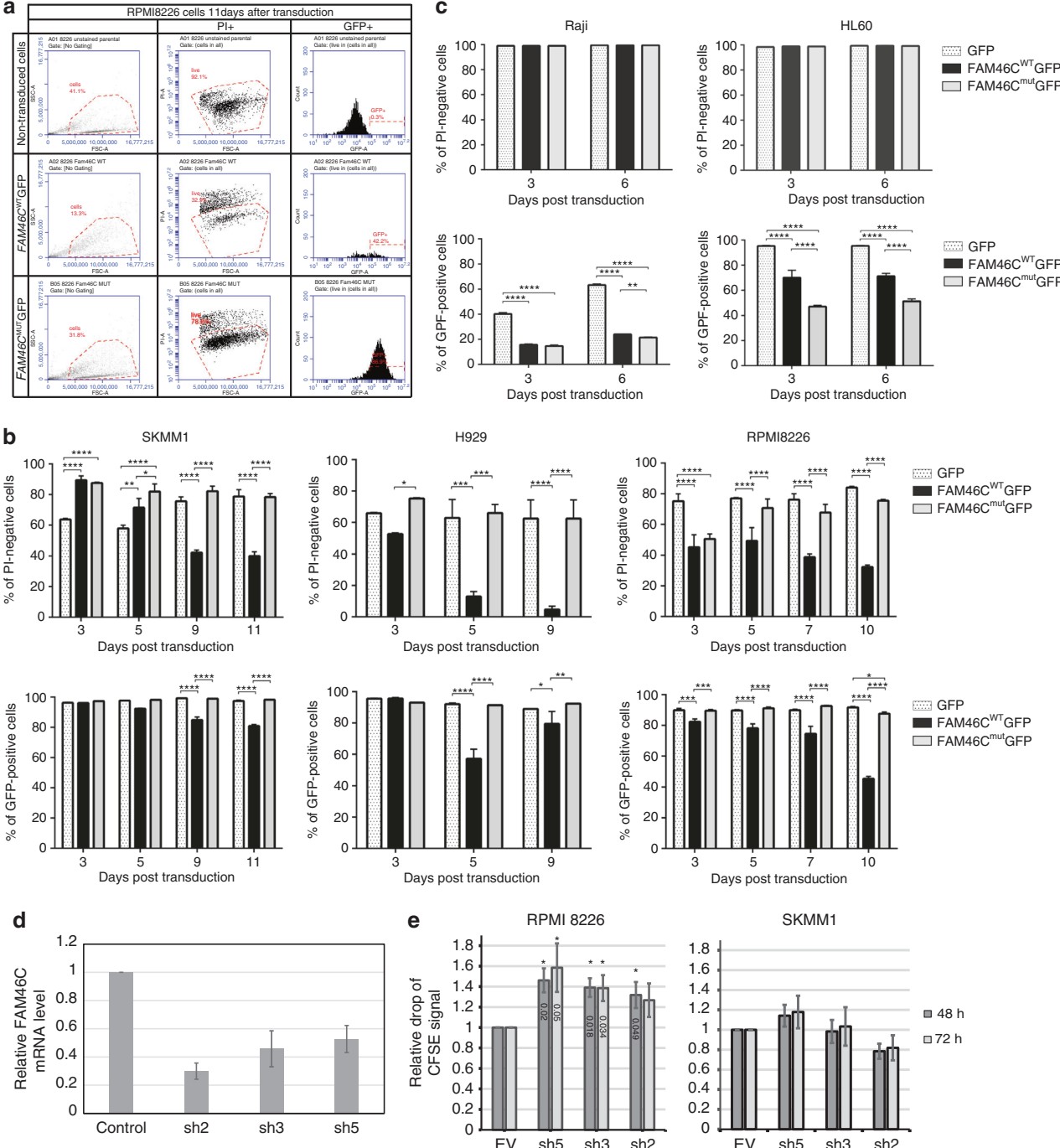

**Fig. 4** FAM46C control survival and proliferation of multiple myeloma cells. **a–c** Expression of *FAM46C*WT induces death in multiple myeloma cells. **a** An example gating strategy for defining transduction efficiency and cell viability. Forward scatter (FSC) and side scatter (SSC) gate were used to separate debris from intact cells. Viability of RPMI8226 cells overexpressing either FAM46CWT-GFP or FAM46Cmut-GFP was analyzed using propidium iodide (PI) staining on the 11th day post-transduction. GFP expression (GFP+) was evaluated in parallel in PI-negative cells. **b, c** Summary of flow cytometry analyses presented as bar graphs showing GFP expression level and reduced viability of multiple myeloma cell lines (SKMM1, H929, and RPMI8226 in **b** and Raji and HL60 in **c** throughout the time course of GFP, FAM46CWT-GFP, and FAM46Cmut-GFP expression. The data are presented as percentage of cells ± SD ($n = 3$). *P* values were calculated using two-way ANOVA tests (*$P < 0.05$, **$P < 0.01$, ***$P < 0.001$, ****$P < 0.0001$). **d, e** shRNA-mediated silencing of *FAM46C* enhances proliferation rate of RPMI8226 MM cell line expressing wild-type protein but not SKMM1 harboring FAM46C mutation. Cells were transduced with lentiviral vectors expressing shRNA and empty vector as control. Stable transduced cell lines were stained with CFSE. Cell division was monitored by flow cytometry after 48 and 72 h by levels of CFSE dilution. **d** Reverse transcription qPCR analysis of the FAM46C silencing efficiency. *Bars* represent mean values ± SD ($n = 3$). **e** The rate of proliferation of shRNA treated cells normalized to the control transduction. *Bars* represent mean values ± SD. *P*-values were calculated using Student's *t*-test (*$P < 0.05$), ($n = 3$)

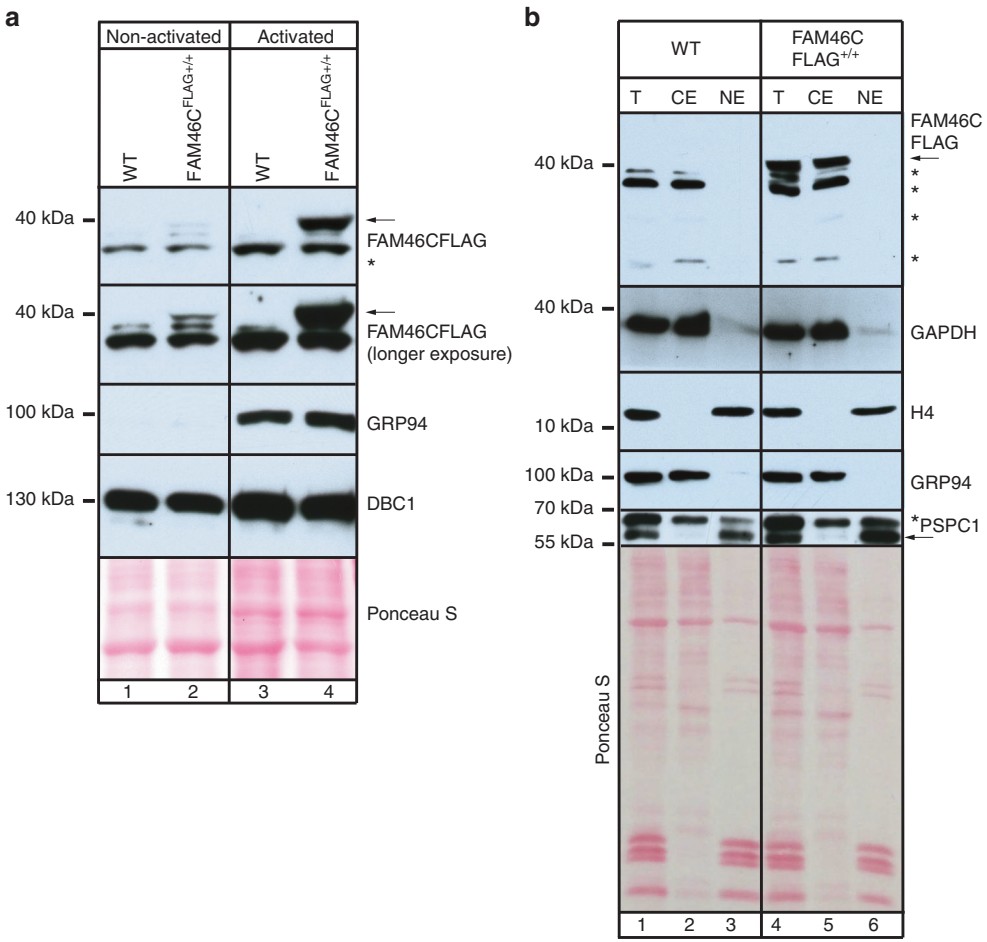

**Fig. 5** FAM46C is induced in activated B lymphocytes and localizes mainly in the cytosol. **a** Splenocytes from *FAM46C-FLAG +/+* knock-in mice and control animals were isolated and then cultured in presence of IL4 and LPS for 72 h. The level of FAM46C-FLAG was checked by western blot in non-activated (lanes 1–2) and activated (lanes 2–4) cells. GRP94 was used as activation marker, while DBC1 was used as a loading control. **b** Endogenous FAM46C localizes mainly in the cytosol. Splenocytes from WT and *FAM46C-FLAG+/+* mice were isolated and fractionated. The whole-cell (T), cytoplasmic (CE), and nuclear (NE) extracts were analyzed by western blot. FAM46C-FLAG was detected using an anti-FLAG antibody. GAPDH and GRP94 were used as a cytosolic markers, while histone H4 and PSPC1 were used as nuclear markers. *Arrow* indicates the position of FAM46C-FLAG, while *asterisks* indicts nonspecific bands detected by antibodies, which were present in samples isolated from both WT and FAM46C-FLAG animals

protein (FLAG-FAM46C^WT, FAM46C^WT-FLAG, GFP-FAM46C^WT, and NHA-FAM46C^WT; Supplementary Fig. 4). We also tested other MM cell lines (MM1.S, MM1.R) and observed the same effects. Importantly, in contrast to *FAM46C^WT*-GFP, expression of *FAM46C^mut*-GFP or GFP only, did not cause cell death and stable MM cell lines were established (Fig. 4a, b). This observation indicate that cytotoxic and cytostatic effects of *FAM46C* over-expression depend on its enzymatic activity. Moreover, over-expression of GFP-tagged nucleotidyltransferases such as canonical (PAPOLA) or non-canonical poly(A) polymerases (POLS or GLD2) in SKMM1 cell lines did not cause strong cytostatic effects (Supplementary Fig. 5), since the toxicity was visible only at early time points post transduction and stable cell lines were derived. This suggests a unique influence of *FAM46C* on MM cell viability.

Finally, in order to determine if *FAM46C* toxicity is restricted to MM cells, other hematological cell lines, including HL60 (promyelocytic leukemia) and Raji (B-cell lymphoma), were transduced with lentiviruses carrying *FAM46C^WT*-GFP, *FAM46C^mut*-GFP, and GFP. Unlike in MM cells, expression of *FAM46C* had no effect on HL60 or Raji cell growth and viability, and these lines were established after cell sorting (Fig. 4c). It was concluded that *FAM46C* expression specifically induces cell death

of MM cells, however, we cannot exclude that reduced toxicity of FAM46C for HL60 and Raji cell is not related to lower expression levels of the transgene.

Furthermore, in order to verify if FAM46C may positively regulate gene expression in MM cells we have conducted tethering assays performed similarly to the one described for HEK293 cells but implemented for SKMM1 and H929 cells. As expected, tethering of FAM46C^WT but not FAM46C^mut in MM cell lines enhanced the expression of the reporter mRNA (Supplementary Fig. 6).

Mutations in *FAM46C* are found only in a fraction of MM cases. In order to check whether the effect of *FAM46C* on cell survival and proliferation is general to MM cells, we silenced *FAM46C* expression in RPMI8226, a MM cell line expressing wild-type *FAM46C*. To this end, we derived stable cell lines transduced with lentiviral vectors expressing three different shRNA targeting *FAM46C*. Although the level of silencing was moderate (~ 50%, Fig. 4d), basically all shRNAs led to an moderately increased proliferation rate of RPMI8226 cell line as measured using CFSE cell tracer (Fig. 4e). Such an effect was not observed in the SKMM1 MM cell line harboring *FAM46C* mutations (Fig. 4e). This result strongly suggests that *FAM46C* suppresses the growth of all MM cells.

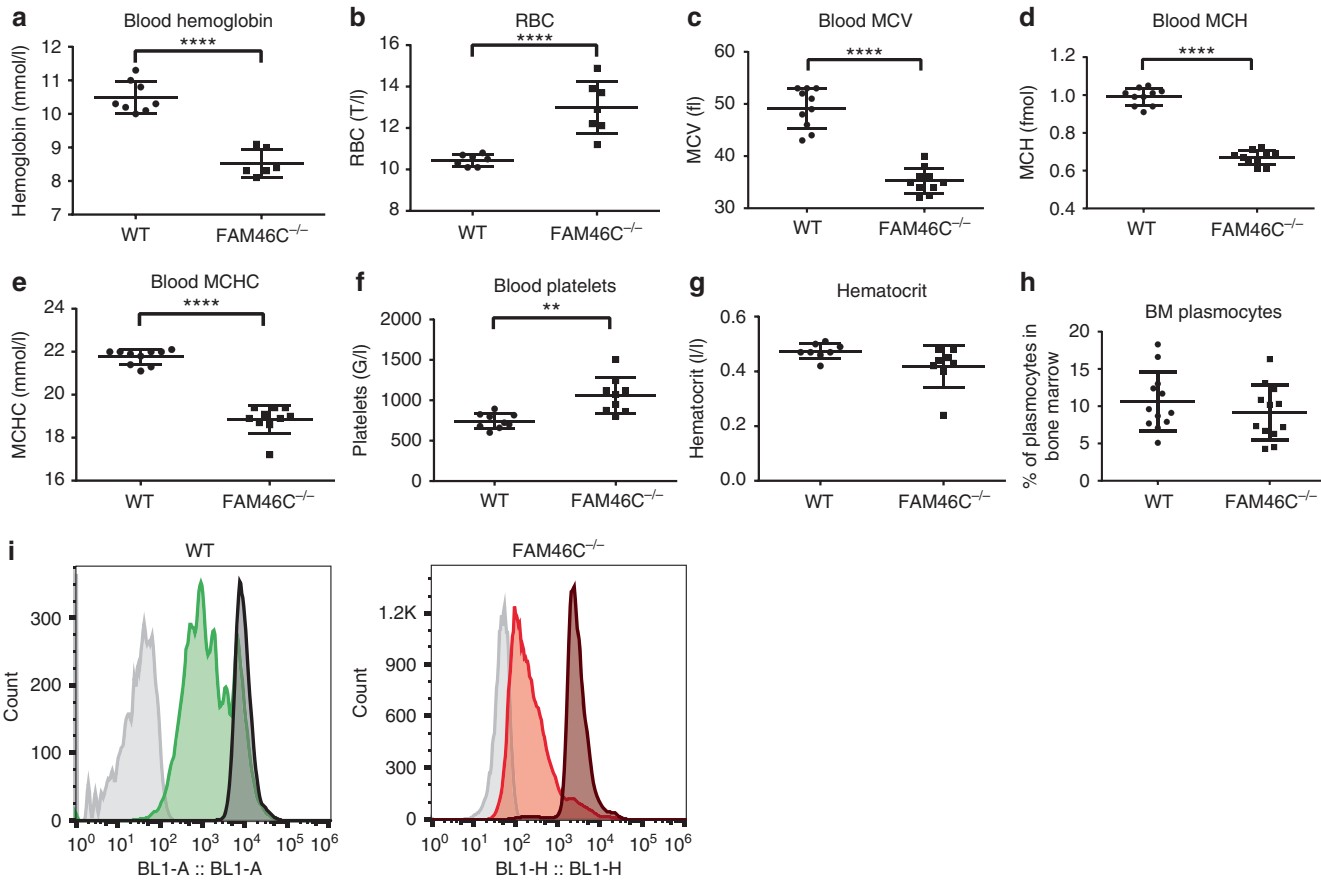

**Fig. 6** *FAM46C* KO 12-week-old mice display reduced blood hemoglobin levels and activated primary B lymphocytes proliferate faster. **a–h** Hematologic parameters of *FAM46C* KO and control animals: **a** Blood hemoglobin concentration (WT, n = 8; *FAM46C* KO, n = 6); **b** red blood cells count (RBC) (WT, n = 7; *FAM46C* KO, n = 7); **c** mean corpuscular volume (MCV) (WT, n = 10; *FAM46C* KO, n = 10); **d** mean corpuscular hemoglobin (MCH) (WT, n = 10; *FAM46C* KO, n = 10); **e** mean corpuscular hemoglobin concentration (MCHC) (WT, n = 10; *FAM46C* KO, n = 10); **f** blood platelets (WT, n = 9; *FAM46C* KO, n = 9); **g** Hematocrit (WT, n = 8; *FAM46C* KO, n = 8); **h** bone marrow (BM) plasmocytes (WT, n = 12; *FAM46C* KO, n = 12). *P*-values were calculated using two-way ANOVA tests (**$P < 0.01$, ****$P < 0.0001$). **i** Analysis of the proliferation rate of activated primary B cells. B lymphocytes isolated from *FAM46C* KO mice and matching WT controls were stained with CFSE and in vitro activated using IL4 and LPS. Cell divisions were monitored by levels of CFSE dilution. One representative experiment of 2 is presented. For each experiment splenic B lymphocytes from three individuals have been pulled, for both WT and mutant mice

**FAM46C KO enhances B lymphocyte proliferation rate**. The analysis of MM cell lines revealed that FAM46C expression affects their growth and survival. Our intensive trials to raise or purchase specific antibodies that recognized FAM46C failed and so we were unable to verify endogenous FAM46C levels in MM cells. Thus, to study the expression of FAM46C in the B-cell lineage, from which MM originates, we have generated a FAM46C-FLAG knock-in mouse using CRISPR/Cas9 methodology. The animals did not display any detectable phenotypes. We have used western blot analysis of IL4 and LPS activated splenocytes and found that FAM46C expression was very strongly enhanced upon activation (Fig. 5a). Furthermore, cell fractionation experiments revealed that it is localized mainly in the cytoplasm (Fig. 5b), which is generally in agreement with our localization studies based on confocal imaging. Such result suggests that FAM46C may act as a non-canonical poly(A) polymerase in the B lymphocyte lineage.

In order to study the potential role for FAM46C in B lymphocytes and more generally in hematopoiesis, we have generated *FAM46C* mutant animals, which harbor a 17 bp deletion and a 301 bp insertion using the CRISPR. This mutation causes a frameshift at position C88 and premature FAM46C protein termination, which takes place 26 amino acids after the

breakpoint. Although, detailed analysis of *FAM46C* KO phenotypes are outside of the scope of this study, the mice did not display any major developmental phenotypes. However, we did detect several hematologic abnormalities (Fig. 6a–h). Homozygous *FAM46C* KO mice (*FAM46C*⁻/⁻) had significantly lower hemoglobin level than age- and sex-matching *FAM46C*⁺/⁺ controls, suggesting that they suffer from anemia. The red blood cells count was also increased while MCV, MCH and MCHC were all significantly decreased. The microcytic and hypochromic erythrocytes imply that insufficiency of hemoglobin production is the most probable cause of the anemia. The MCV/RBC ratio (Mentzer index) is lower in *FAM46C*⁻/⁻ (2.52) than in *FAM46C*⁺/⁺ (4.39), suggesting the anemia is a consequence of a block in globin synthesis rather than iron insufficiency. This hypothesis has to be further tested experimentally. These results are generally in agreement with the unpublished work from the Fleming laboratory described in the Meng Tian PhD thesis, but runs contrary to the Mouse Phenotypic Consortium, which has found *FAM46C* an essential gene. We suppose, that this discrepancy may be due to some linked lethal mutation generated by the consortium.

Importantly, analysis of in culture-activated primary splenic B lymphocytes revealed that *FAM46C* mutation leads to an

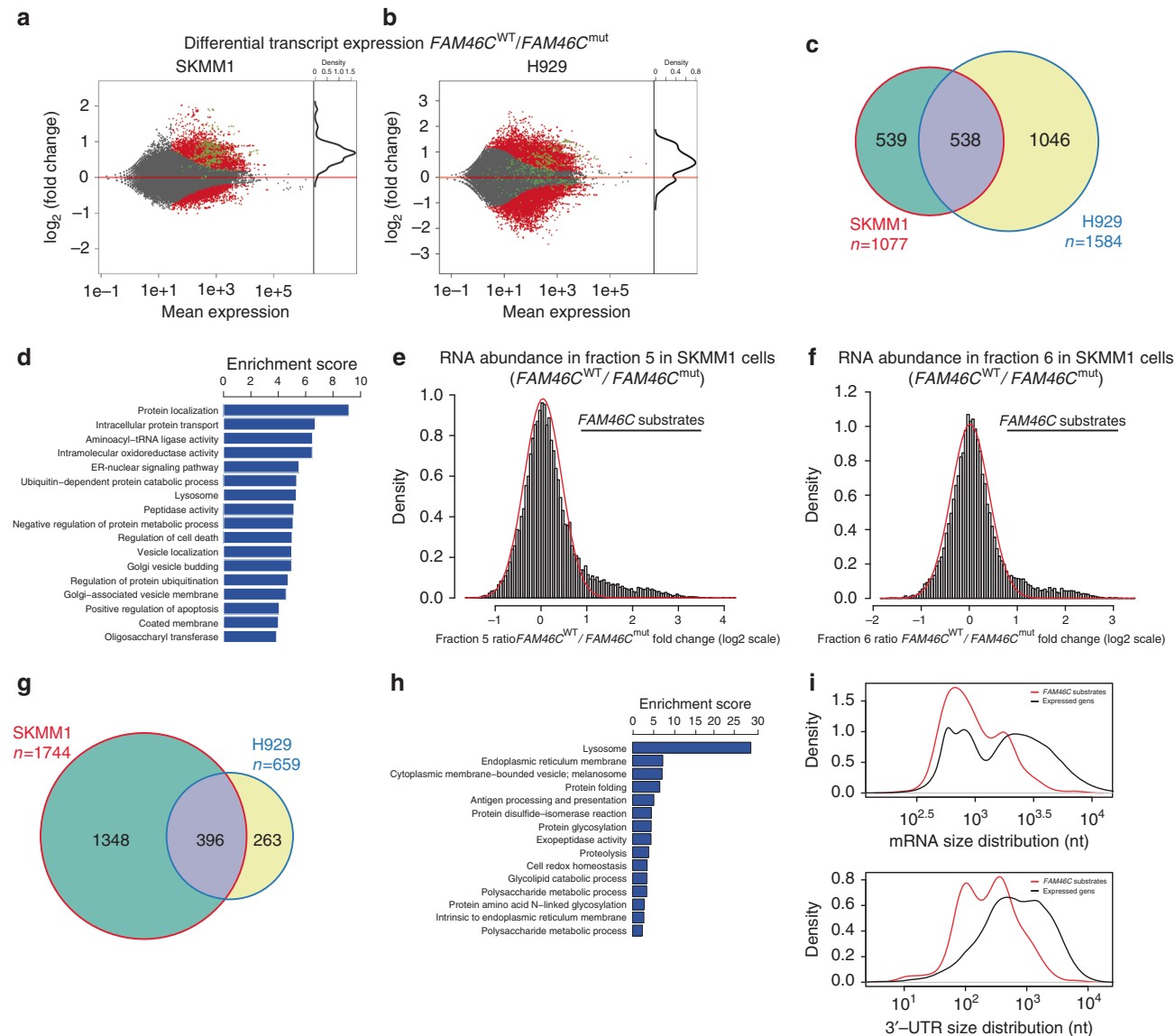

**Fig. 7** Global analysis of FAM46C substrates in MM cells. **a**, **b** MAplots of total RNA sequencing results representing differential expression analysis of FAM46C$^{WT}$-GFP and FAM46C$^{mut}$-GFP-transduced MM cell lines. SKMM1 (**a**) and H929 (**b**) show transcriptome wide moderate deregulation of gene expression. Statistically significant values (FDR < 1%) are shown in *red*. The 396 transcripts constituting the common part for SKMM1 and H929 MM cell lines are marked with *green circles*. **c** Venn diagram demonstrating the overlap between upregulated transcripts in SKMM1 and H929 MM cell lines. **d** Functional GO terms annotation clustering of significantly upregulated transcripts in both MM cell lines. **e**, **f** Histogram representing the distribution of FAM46C-dependent polyadenylation rations monitored in fraction #5 (**e**) and #6 (**f**) in SKMM1 cell lines. Modeled normal distribution is fitted as a *red line* to emphasize the outlying population of FAM46C polyadenylated transcripts. **g** Venn diagram demonstrating the overlap between transcripts shifting to both long poly(A) fractions in FAM46$^{WT}$ overexpressing cells mutually in SKMM1 and H929 MM cell lines. **h** 396 transcripts constituting the common part for SKMM1 and H929 MM cell lines were used in DAVID Functional Annotation Clustering Analysis. **i** The same set of transcripts was used to characterize the 3′-UTRs and total mRNA lengths showing a bias towards shorter species in either

increased proliferation rate, what indicates that *FAM46C* decreases the growth rate of B cells (Fig. 6i). In conclusion, *FAM46C* acts as a growth suppressor not only for MM cells but also more generally in B-cell lineage cells.

**FAM46C polyadenylates multiple ER-targeted protein mRNAs.** In order to identify *FAM46C* substrates in MM cells, deep sequencing of total RNA from SKMM1 and H929 MM cell lines was performed after 3 days of *FAM46C*$^{WT}$-GFP and *FAM46C*$^{mut}$-GFP overexpression. RNA samples were prepared in triplicates or quadruplicates and depleted of rRNA, and strand-specific total RNA libraries were prepared. The samples were sequenced

and mapped to an average depth of 15 million uniquely aligning reads. Differential expression analyses revealed significant differences in transcriptomes and responses to *FAM46C* expression among tested MM cell lines (Fig. 7a, b and Supplementary Data 1). When *FAM46C* was overexpressed, H929 cells had an increase in steady-state levels of many transcripts that encode proteins involved in the interferon response. This effect was not observed in the SKMM1 cell line. Nevertheless, there was a significant overlap (538 mRNAs) between upregulated transcripts among MM cell lines (Fig. 7c). Notably, functional annotation clustering analysis of upregulated transcripts in SKMM1 and H929 MM cells expressing *FAM46C*$^{WT}$ (FDR < 1%), (using

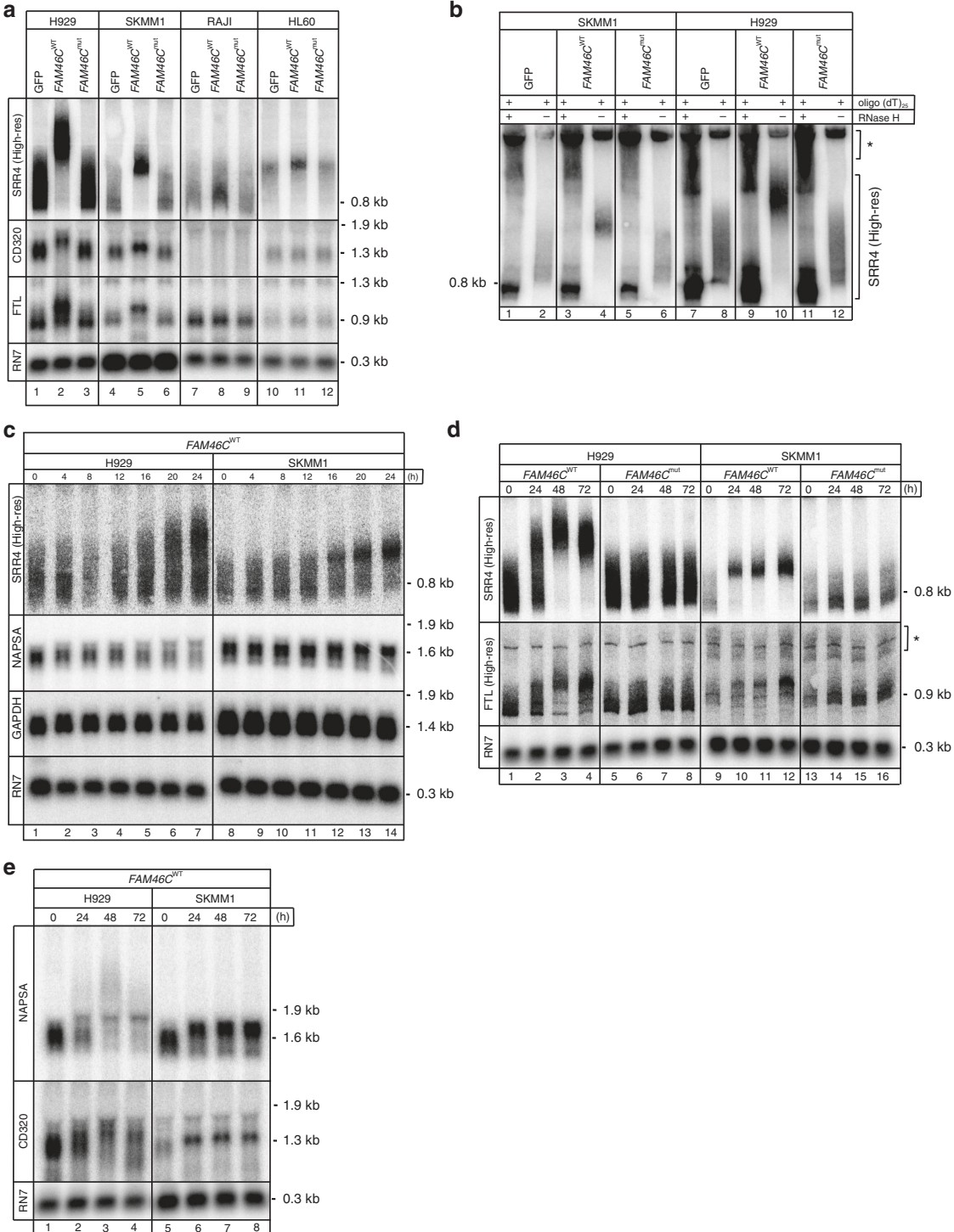

**Fig. 8** FAM46C expression results in polyadenylation of selected mRNAs in MM cells. **a** Northern blot analysis of *SSR4*, *CD320*, and *FTL* transcripts from H929 (lanes 1–3), SKMM1 (lanes 4–6), Raji (lanes 7–9) and HL60 (lanes 10–12) cells transduced with GFP (lanes 1, 4, 7, 10), FAM46C$^{WT}$-GFP (lanes 2, 5, 8, 11), and FAM46C$^{mut}$-GFP (lanes 3, 6, 9, 12). *Asterisks* indicate cross-hybridization signals. **b** The *SSR4* transcript is extensively polyadenylated by FAM46C in MM cells. High-resolution northern blot analysis of *SSR4* transcripts from SKMM1 (lanes 1–6) and H929 (lanes 7–12) transduced with GFP (lanes 1, 2, 7, 8), FAM46C$^{WT}$-GFP (lanes 3, 4, 9, 10), and FAM46C$^{mut}$-GFP (lanes 5, 6, 11, 12) after RNase H treatment (lanes 1, 3, 5, 7, 9, 11) to remove the poly(A) tail in presence of oligo(dT)$_{25}$. Control reactions were carried out in the presence of oligo(dT) without RNase H (lanes 2, 4, 6, 8, 10, 12). Kinetics of polyadenylation of *SSR4*, *NAPSA*, *FTL*, and *CD320* transcripts over the time course of *FAM46C* expression. **c** High- and low-resolution northern blot analysis of *SSR4*, *NAPSA*, *GAPDH*, and *RN7* transcripts from H929 (lanes 1–7) and SKMM1 cells (lanes 8–14) transduced with FAM46C$^{WT}$-GFP up to 24 h. **d** High-resolution northern blot analysis of *SSR4* and *FTL* transcripts from H929 (lanes 1–8) and SKMM1 cells (lanes 9–16) transduced with FAM46C$^{WT}$-GFP (lanes 1–4, 9–12) and FAM46C$^{mut}$-GFP (lanes 5–8, 13–16) up to 72 h. **e** Northern blot analysis of *NAPSA* and *CD320* transcripts from H929 (lanes 1–4) and SKMM1 cells (lanes 5–8) transduced with FAM46C$^{WT}$-GFP up to 72 h

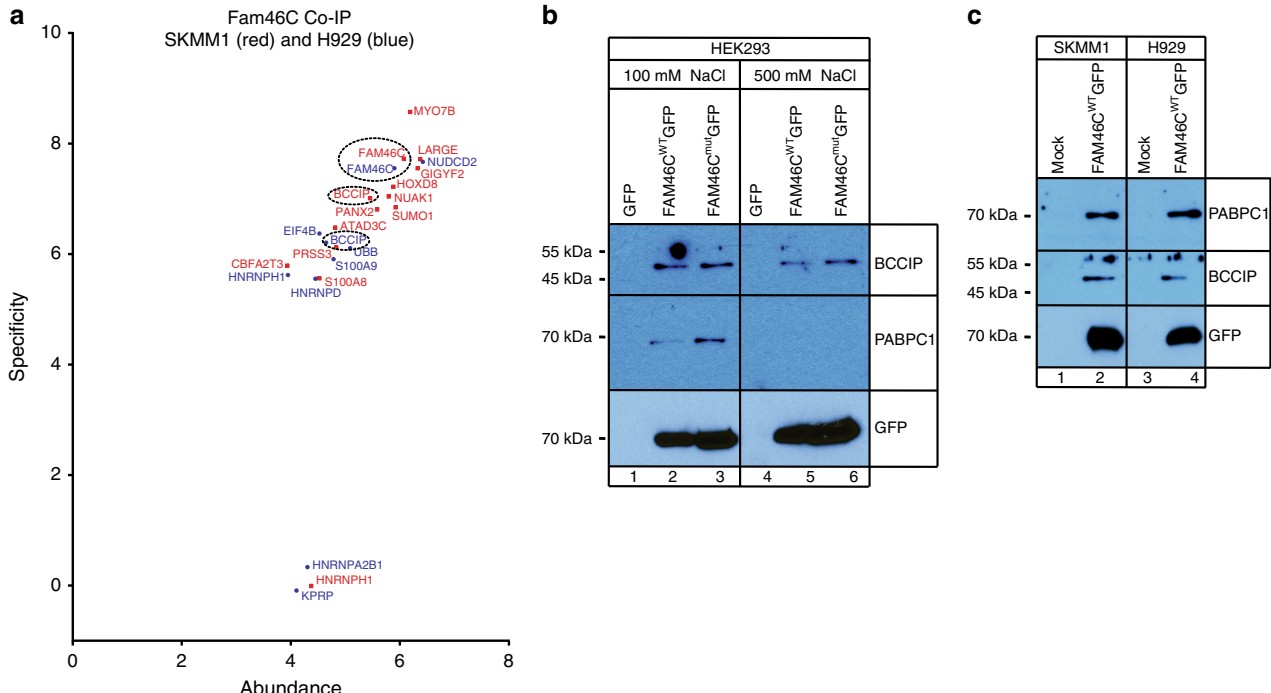

**Fig. 9** FAM46C interacts with BCCIPβ and PABPC1 in human cells. **a** Visualizations of Co-IP-MS experiments using GFP-tagged FAM46C expressed in myeloma cells H929 (shown in *blue*) and SKMM1 (shown in *red*). Estimated quantities of identified proteins were calculated using the label-free quantification (LFQ) algorithm and are represented as dot–plot graphs. Protein abundance was calculated as LFQ intensity of the protein signal divided by its molecular weight and is shown on the x axis on a logarithmic scale. Specificity was defined as the ratio of protein LFQ intensity measured in the bait Co-IP to the background level (protein LFQ intensity in a sample) and is shown on the y axis on a logarithmic scale. **b**, **c** Western-blot detection of BCCIPβ and PABPC1 co-immunoprecipitated with: **b** FAM46C$^{WT}$-GFP and FAM46C$^{mut}$-GFP from stable HEK293 cell lines; **c** FAM46C$^{WT}$-GFP from transduced SKMM1 and H929 MM cell lines. Purification was performed at low (lanes 1–3) and high (lanes 4–6) ionic strengths. GFP-tagged FAM46C was detected with α-GFP antibodies. Please note that the lack of the GFP signal in the GFP only control (lanes 1 and 4) is because the blot was cutoff to avoid very strong GFP signal, which could influence the readout

DAVID) revealed significant enrichment towards transcripts encoding proteins that pass through ER/Golgi and are transported into lysosomes or exported outside the cell (Fig. 7d and Supplementary Fig. 7a, b).

In order to take a closer look at changes after wild-type *FAM46C* expression, we fractionated mRNA based on poly(A) tail length from H929 and SKMM1 MM cells expressing *FAM46C*$^{WT}$-GFP, *FAM46C*$^{mut}$-GFP, and GFP for 72 h. RNA-seq libraries were prepared from two RNA fractions with the longest tails (fractions 5 and 6) and one fraction with the shortest poly(A) tail (fraction 1), followed by library preparation, sequencing, and initial analysis (analogous to the one described for RNA-seq).

In order to identify FAM46C substrates, we aimed to determine differences in the enrichment of mRNA species in the highly polyadenylated fractions in cells transduced with wild-type *FAM46C* compared to cells transduced with the catalytically inactive mutant. To this end, we calculated "polyadenylation ratios" by dividing the levels of individual mRNA in long poly(A) fractions (5 and 6) relative to the short fraction 1 in *FAM46C*$^{WT}$ cells using relative levels observed in *FAM46C*$^{mut}$ cells. Plotting the histograms of polyadenylation ratios revealed a significant set of transcripts with elongated poly(A) tails, which did not fit the normal distribution (Fig. 7e, f and Supplementary Fig. 7c, d). This approach allowed us to identify 1744 mRNAs for SKMM1 and 659 mRNAs for H929 that shift towards longer poly(A) fractions in response to forced *FAM46C*$^{WT}$ expression; thus, they are presumably polyadenylated by this protein. The top-scoring transcripts were compared and as much as 60% of all H929 statistically significant hits were in common with SKMM1

hits, suggesting strong substrate specificity of *FAM46C* (Fig. 7g and Supplementary Data 2). Moreover, in the case of SKMM1 cell line the levels of basically all polyadenylated transcripts were increased confirming the role of *FAM46C*-mediated polyadenylation in enhancing mRNA stability. The correlation between the polyadenylation and mRNA levels was also visible for H929 MM cells. The effect, was, however, less pronounced, most probably due to the interferon response observed in H929—but not in SKMM1—which is known to have a strong influence on transcriptome hemostasis.

Importantly, clustered GO-term analysis of transcripts polyadenylated by *FAM46C* using DAVID (https://david.ncifcrf.gov/) revealed that, as in the case of standard RNA-seq analysis, many *FAM46C* substrates encode proteins that pass through ER/Golgi and are transported into lysosomes or exported outside the cell (Fig. 7h)[21, 22]. Moreover, despite filtering for highly-expressed transcripts in an analysis that usually creates a bias towards longer mRNAs, we found that short mRNAs with short 3′-UTRs are more prone to being substrates of *FAM46C* (Fig. 7i). Sequence analysis of 3′-UTRs of *FAM46C*-polyadenylated mRNA revealed that, although not a single specific-sequence motif could be identified, there was a striking enrichment of highly statistically significant GpU- and CpU-rich motifs compared to 3′-UTRs of all expressed genes (Supplementary Fig. 7e). These motifs or secondary structures may be recognized by *FAM46C* directly or indirectly through other recruiting factors.

*SSR4* (signal sequence receptor subunit delta), which is involved in the translocation of proteins across the endoplasmic reticulum membrane, was identified as a top hit *FAM46C* substrate in both MM cell lines. SSR4, as well as other

ER-targeted proteins, such as *CD320*, *NAPSA*, and *FTL*, were selected for further validation analysis based on the fact that ER stress is considered to be one of the most important processes in maintaining and targeting MM cells and augmentation of ER overload by proteasome inhibitors has emerged as an important therapeutic strategy for MM treatment.

Northern blot analysis confirmed that *SSR4*, *CD320*, and *FTL* transcripts migrate much slower, as their poly(A) tails are extended in H929 and SKMM1 MM cells expressing *FAM46C*[WT]-GFP (Fig. 8a); this effect was not observed when GFP only or *FAM46*[mut]-GFP were expressed. Moreover, poly(A) tails were not extended in other cell lines derived from hematological malignancies, such as in Raji and HL60 transduced with *FAM46C*. Furthermore, *SSR4* mRNA is polyadenylated at the 3′ end as revealed by analysis using RNase H cleavage assay with oligo(dT)25 followed by northern blot (Fig. 8b); similar molecular phenotypes were determined for the translocon-associated protein *SSR2* (Signal Sequence Receptor, Beta) (Supplementary Fig. 8a, b).

Additionally, poly(A) tail length dynamics of selected mRNAs during *FAM46C* expression were also evaluated in a time course. Cells were transduced with *FAM46*[WT]-GFP and *FAM46C*[mut]-GFP, collected at indicated time points, and analyzed using northern blot. The first extended molecules were detected after 16 h of *FAM46C*[WT]-GFP expression in both cell lines (Fig. 8c–e). In order to determine if the effect of *SSR4* mRNA polyadenylation was specific to the FAM46C protein, we constructed lentiviral vectors allowing for expression of other RNA poly(A) polymerases, including GLD2-GFP, PAPOLA-GFP, and POLS-GFP, and transduced H929 and SKMM1 cells, and confirmed that this phenotype is caused by expression of *FAM46C* gene (Supplementary Fig. 8c). It was concluded that *FAM46C* polyadenylates a broad spectrum of substrates in MM cell lines, inducing strong enrichment in those encoding ER-targeted proteins.

**FAM46C is a monomer and co-purifies with BCCIPβ and PABPC1.** Cytoplasmic non-canonical poly(A) polymerases usually obtain specificity through associated RNA-binding proteins[23, 24]. In order to identify the proteins interacting with *FAM46C*, co-immunoprecipitation (Co-IP) experiments were performed using SKMM1 and H929 MM cells expressing *FAM46C*[WT]-GFP. High-resolution mass spectrometry (MS) followed by label-free quantification (LFQ) revealed that BCCIPβ is the only protein enriched in both MM cell lines (Fig. 9a, Supplementary Data 3). Analyses performed in HEK293 cells also allowed for the identification of cytoplasmic poly(A)-binding protein PABPC1 as a potential FAM46C interactor (Fig. 9b, Supplementary Data 3). PABPC1 was an interesting target due to the poly(A) polymerase enzymatic activity of *FAM46C*. Interactions between FAM46C and BCCIPβ, PABPC1 were confirmed by Co-IPs followed by western blot analyses of HEK293, SKMM1, and H929 MM stable cell lines expressing FAM46C[WT]-GFP. Interaction of FAM46C with BCCIPβ was very stable under all experimental conditions tested; however, interaction with PABPC1 was identified only at physiological salt concentration (Fig. 9b). Despite the fact that HEK293 cells express both FAM46A and FAM46B proteins, none of these co-purified with FAM46C, suggesting that they do not form heteromers.

PABPC1 and BCCIPβ proteins were selected for functional evaluation as potential FAM46C interactors. First, it was determined whether these proteins had any impact on mRNA after tethering to an RL reporter construct. Similarly to previous experiments, cells were transiently transfected with pRL-5BoxB, pNHA-BCCIPβ, and pNHA-PABPC1 plasmids and, after 24 h, RL mRNA and protein levels were analyzed. BCCIPβ did not

affect mRNA; however, PABPC1 tethering led to increased RL mRNA and protein levels (Supplementary Fig. 9a–c). Next, we examined FAM46C poly(A) polymerase activity in cells with reduced BCCIP and PABPC1 levels after siRNA treatment. HEK293 cells were treated with siRNA against *BCCIP* or *PABPC1* mRNAs and, after 3 days, reseeded and co-transfected with pRL-5BoxB, pNHA-FAM46C, and pNHA-FAM46C[mut]. BCCIPβ and PABPC1 protein levels after siRNA treatment were determined by western blot analysis (Supplementary Fig. 9d). After 24 h, steady state levels of reporter mRNA and protein were analyzed (Supplementary Fig. 9e). In all cases, FAM46C tethering led to increases in steady state RL mRNA and protein levels (Supplementary Fig. 9e). In conclusion, FAM46C does not form stable macromolecular assemblies and interacting proteins have no direct effect on its poly(A) polymerase activity. Thus, the mechanism through which FAM46C gains specificity towards particular mRNAs remains unknown.

**Discussion**

Template-independent elongation of the 3′-end of mRNA by nucleotidyltransferases, known as poly(A) polymerases, plays a crucial role in RNA processing that affects mRNA stability and translational efficiency. Here we provide strong in vitro and in culture evidence demonstrating that the *FAM46C* gene encodes a novel poly(A) polymerase that acts as an B-cell lineage growth suppressor. Reintroduction of *FAM46C* into MM cell lines with dysfunctional endogenous gene leads to cell death while its silencing in cells expressing WT proteins enhances proliferation rate. Furthermore, activated primary B lymphocytes isolated from *FAM46C* KO animals divide faster than the wild type, suggesting that FAM46C acts more generally in the B lymphocyte lineage. Hematologic phenotypes of *FAM46C* KO animals, in which red blood cell and platelet counts are increased further suggests that this enzyme controls proliferation of other blood cells or common progenitor. To summarize, we are strongly confident that MM cells must benefit from mutations in FAM46C, however we cannot exclude that FAM46C suppresses the growth of other cell types than ones derived from the B-cell linage. Recent studies from the Xin Lab suggest that FAM46C can also be involved in regulation of proliferation of hepatocellular carcinoma cells[25, 26].

Our data based on SKMM1 and H929 cell lines strongly indicate that *FAM46C* has a relatively broad spectrum of substrates with significant enrichment towards ER/Golgi-targeted proteins, which may explain the toxicity to MM cells since ER and Golgi apparatus homeostasis is essential for survival. Although we did not determine the specificity of the FAM46C protein for particular substrates, we found that the toxicity of *FAM46C* to MM cells is unique, since overexpression of other poly(A) polymerases, such as GLD2 and canonical PAP, was not harmful to MM cells. Similarly to the overexpression of the wild-type form of *FAM46C* in HL60, there was no significant effect on the survival of Raji and 293 T cell lines. These results suggest that the toxicity of *FAM46C* to MM cell lines is related to the expression of ER/Golgi-targeted proteins, which may result in ER overload and induction of an unfolded protein stress response in these cells[27, 28]. However, how exactly the numerous genetic differences between cancer cell lines makes them sensitive to *FAM46C* expression remains to be established.

Importantly, this work provides the first experimental evidence directly linking RNA 3′ end polyadenylation process with cell proliferation, notably in the B-cell lineage implying the possible role of FAM46C mutations as driver mutations in MM. Interestingly, a well-established regulator of polyadenylation in gametogenesis, cytoplasmic polyadenylation element binding (CPEB), is also implicated in senescence and was suggested to be a tumor

**Table 1 List of human non-canonical PAPs and PUPs**

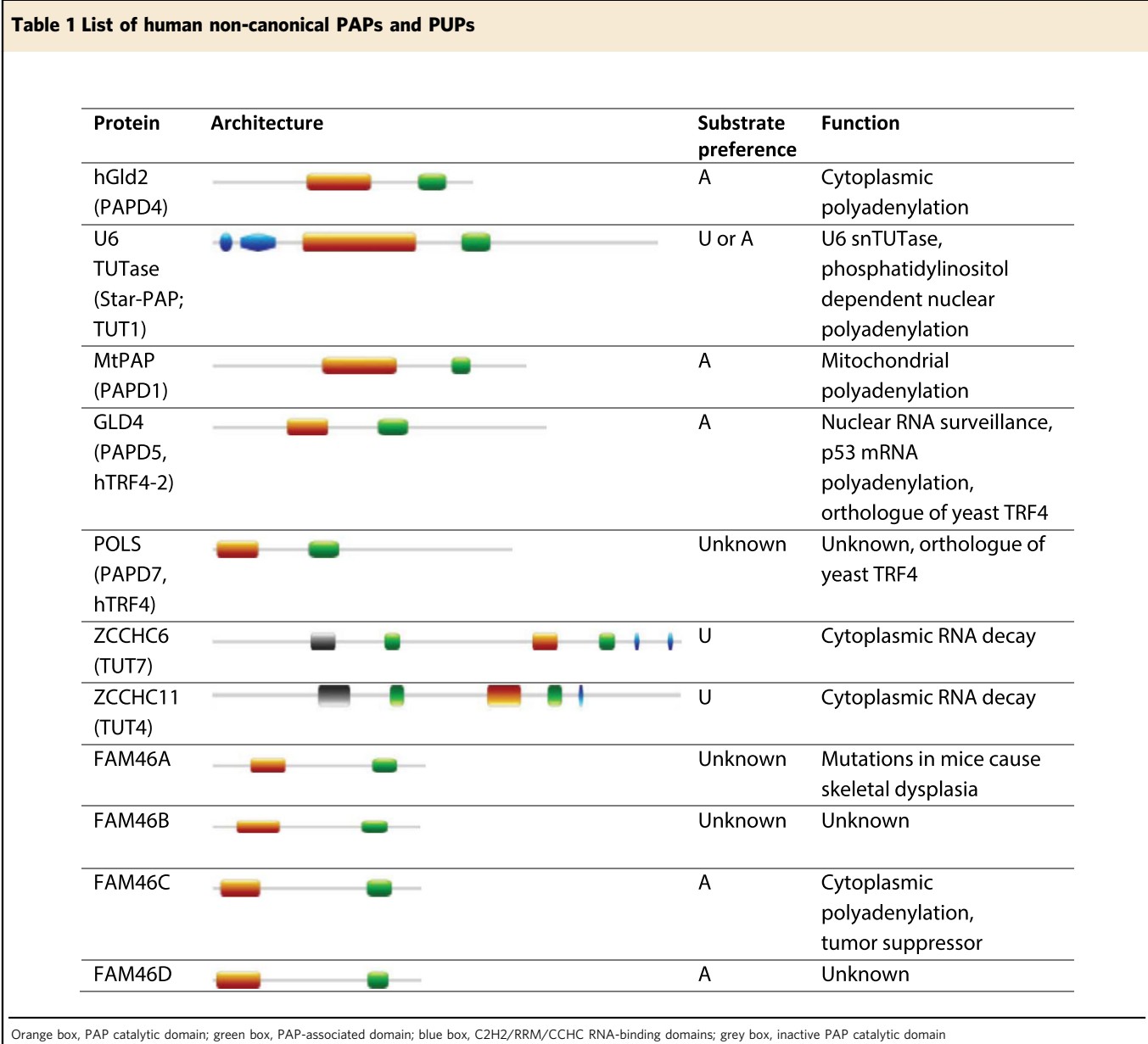

| Protein | Architecture | Substrate preference | Function |
|---|---|---|---|
| hGld2 (PAPD4) | | A | Cytoplasmic polyadenylation |
| U6 TUTase (Star-PAP; TUT1) | | U or A | U6 snTUTase, phosphatidylinositol dependent nuclear polyadenylation |
| MtPAP (PAPD1) | | A | Mitochondrial polyadenylation |
| GLD4 (PAPD5, hTRF4-2) | | A | Nuclear RNA surveillance, p53 mRNA polyadenylation, orthologue of yeast TRF4 |
| POLS (PAPD7, hTRF4) | | Unknown | Unknown, orthologue of yeast TRF4 |
| ZCCHC6 (TUT7) | | U | Cytoplasmic RNA decay |
| ZCCHC11 (TUT4) | | U | Cytoplasmic RNA decay |
| FAM46A | | Unknown | Mutations in mice cause skeletal dysplasia |
| FAM46B | | Unknown | Unknown |
| FAM46C | | A | Cytoplasmic polyadenylation, tumor suppressor |
| FAM46D | | A | Unknown |

Orange box, PAP catalytic domain; green box, PAP-associated domain; blue box, C2H2/RRM/CCHC RNA-binding domains; grey box, inactive PAP catalytic domain

suppressor[29–31]. However, the enzyme responsible for CPEB-dependent polyadenylation in somatic cells remains unknown, as knock-out of the poly(A) polymerase GLD2, which cooperates with CPEB in *Xenopus* oocytes[32, 33], has no visible phenotype in mice[34].

PABPC1 and BCCIP proteins are novel FAM46C interactors that had not been reported in high throughput screens. The cytoplasmic-nuclear shuttling protein PABPC1 is a *FAM46C* interactor and is directly related to its physiological function, since it binds to nascent poly(A) tracts, affecting initiation of protein translation and both RNA processing and stability; however, the additional factors determining *FAM46C* substrate specificity require further investigation. Furthermore, FAM46C displays very weak distributive activity in vitro, while it seems to be more active in vivo, thus additional factors responsible for its optimal activity probably remain to be identified.

Our identification of FAM46 proteins as active poly(A) polymerases suggests that the human genome encodes at least 11 putative ncPAPs and PUPs (4 FAM46 family members and 7

other previously annotated genes; Table 1). FAM46C, as well as all other members in this family, may positively regulate gene expression in various human tissues and organs, since they are differentially expressed during development and several tissues express multiple FAM46 proteins at the same time[35]. Further research is necessary in order to determine the physiological roles of this novel family of poly(A) polymerases. Data obtained thus far strongly suggest that gene expression regulation by cytoplasmic polyadenylation may play a role in various other cellular processes. Our initial analysis of *FAM46C* KO animals suggests several hematologic abnormalities, like anemia, what is in agreement with previously reported data. However, how exactly FAM46C affects erythropoiesis remains to be established.

Recent studies have shown that loss-of-function mutation within *FAM46A* is associated with bone abnormalities in mice and in autosomal recessive retinitis pigmentosa in humans; however, its molecular function remains unexplained[6, 12, 36, 37]. *FAM46A* is expressed in the ameloblast nuclei of tooth germs and may be important for the formation of enamel in teeth[38].

The *FAM46A* sequence polymorphisms SNP (rs11040) and VNTR, within its coding sequence in exon 2 (15bp VNTR repeats may vary from two to seven repeats), were also associated with a susceptibility to tuberculosis[39]. *FAM46A* expression has been reported to increase over two-fold in macrophages after stimulation by heat-attenuated *Mycobacterium tuberculosis* (H37Rv). The VNTR polymorphism in the *FAM46A* gene has also been associated with susceptibility to large-joint osteoarthritis (hip and knee) and non-small cell lung cancer[40, 41]. *FAM46D* expression is strongly upregulated in transgenic mice (MALTT) that exhibit autism spectrum disorder-like behaviors (ASD)[42]. Since FAM46 proteins are active poly(A) polymerases, these data strongly suggest that additional phenotypes may be related to the regulation of posttranscriptional gene expression.

In conclusion, this study identified that the human *FAM46C* gene encodes a novel poly(A) polymerase and showed FAM46C potential to regulate gene expression posttranscriptionaly resulting in an significant impact on several cell types in the hematopoetic lineages in mice. Thus, demonstrating importance of RNA poly(A) tail homeostasis for cancer cell survival which could be further pursued as a potential therapeutic target for MM treatment.

## Methods

**Cell lines**. MM cell line SKMM1 was kindly provided by W. Michael Kuehl, M.D., Genetics Branch Head, Molecular Pathogenesis of Myeloma Section National Cancer Institute), H929 (ATCC; CRL-9068), MM1.S (ATCC; CRL-2974), MM1.R (ATCC; CRL-2975), RPMI8226 (ATCC; CCL-155), and Raji Burkitt's lymphoma (ATCC; 85011429) were cultured in RPMI 1640 ATCC's modified (Invitrogen) supplemented with 10% FBS (Invitrogen) and penicillin/streptomycin (Sigma; P4083). HL60 cells were cultured in IMDM (Invitrogen) with 20% fetal bovine serum (FBS) and penicillin/streptomycin. HEK293T (ATCC; CRL-3216), Flp-In-293 Cell Line (Invitrogen; R78007), HeLa (ATCC; CCL-2), and U2OS (ATCC; HTB-9) were cultured in Dulbecco's modified Eagle's medium (DMEM; Invitrogen) supplemented with 10% FBS (Gibco) and penicillin/streptomycin at 37 °C in 5% $CO_2$. All cell lines were tested for mycoplasma contaminations upon arrival and routinely during their culture.

Induction of stable cell lines was obtained by addition of tetracycline (Gibco) to a final concentration of 100 ng/ml. siRNAs and plasmid transfections were performed with Lipofectamine 2000 and Opti-MEM media (Invitrogen) according to the manufacturer's instructions. For transcription inhibition, actinomycin D (Sigma; A9415) was added to a final concentration of 5 µg/ml at the times indicated.

**MM cell line genotyping**. The exon 2 of FAM46C gene was amplified using hFAM46C_ex2_1F and hFAM46C_ex2_1R primers, which were designed with Oligo software (MBI, USA) and Primer-Blast[43]. Purified PCR products were sequenced and then thoroughly analyzed with Mutation Surveyor 4.0 software (SoftGenetics) against the presence of mutations and other differences compared to GenBank reference sequences NC_000013 and NC_000001.11. The Mutation Surveyor allowed for detection of minor alleles down to 5%, basing analysis of both strands. Sequencing results are summarized in Supplementary Table 1.

**General molecular biology techniques**. General molecular biology techniques were conducted as described in Molecular Cloning[44]. Low-molecular weight RNA samples were separated on 4–6% acrylamide gels containing 7M urea in 0.5× TBE buffer and transferred to a Hybond N+ membrane by electrotransfer in 0.5× TBE buffer. High-molecular weight RNA samples were separated on 1.2% agarose gels containing formaldehyde in 1× NBC buffer and transferred to membranes by capillary elution using 8× SSC buffer. After transfer membranes were stained with 0.03% methylene blue in 0.3 M NaAc pH 5.3 for 10' at room temperature, scanned and then destined with water. RNA was immobilized on membranes by 254 nm UV light using a UVP CL-1000 crosslinker. Radioactive probes were prepared with a DECAprime II DNA Labeling Kit (Invitrogen). Northern blots were carried out in PerfectHyb Plus Hybridization Buffer (Sigma), scanned with Fuji Typhoon FLA 7000 (GE Healthcare Life Sciences) and analyzed with Multi Gauge software Ver. 2.0 (FUJI FILM). All western blotting experiments were performed in accordance with the technical recommendations of the antibodies' suppliers. All antibodies used in this study are listed in Supplementary Table 2. Unprocessed scans of selected major blots and gels are shown in the Supplementary Fig. 10.

**Cloning and construct preparation**. The primers used for cloning are listed in the Supplementary Data 4. All plasmid-encoding constructs for tethering assays and

miRNA were cloned using classical restriction enzyme digestion and ligation approaches; all other constructs were prepared with SLIC[45, 46]. The pClneo-NHA vector and pRL-5BoxB (Renilla luciferase containing five BoxB structures) were kindly provided by Prof. Witold Filipowicz (FMI). pNHA-GENE constructs were prepared using classical restriction ligation methods. All pcDNA-GENE-TEV-eGFP constructs were prepared by PCR amplification from human cDNA using the appropriate primers listed in Supplementary Data 4. PCR products were subsequently cloned into a modified version of the pcDNA™5/FRT/TO vector (Invitrogen) using the SLIC method. Plasmid pcDNAFAM46CmutGFP was obtained by site-directed mutagenesis using the pcDNAFAM46CmutF and pcDNA-FAM46C$^{WT}$GFP plasmid served as a template; final constructs were verified by sequencing. The HA-tag was inserted into pKK-NoTag made in PCR reaction with primers N_Met-HA_for and N_Met-HA_rev using Acc65I and BamHI cut pKK-NoTag and then assembled with SLIC approach giving pKK-HA. pKK-NoTag is a modified version of pcDNA5FRTTO (All additional information will be described in forthcoming publication). pKK-HA-FAM46C construct was cloned using classical restriction ligation protocol into pKK-HA BshTI and NheI restriction sites. To obtain pCI-NHAFAM46DWT and pCI-NHAFAM46Dmut PCR product with primers NHFAM46DF and NHFAM46DR and pKK-FAM46DWT or pKK-FAM46Dmut plasmids, respectively and cloned into EcoRI and NotI sites in pCI-NHA vector. All SFFV (pLVX) lentiviral constructs were prepared in modified HIVSFFVRFP plasmids, which was kindly provided by Dr Verhoeyen Els (ENS de Lyon). All GENE-TEV-eGFP inserts were obtained from a PCR reaction with the appropriate plasmid pcDNA-GENE-TEV-eGFP and universal SFFV_F and SFFV_R primers. PCR products were cloned into the plasmid HIVSFFVRFP and digested by BamHI and XhoI using the SLIC protocol. To make HIVSFFV-NHA-FAM46C construct, PCR product with primers 15_129_F and 15_129_R and pCI-NHAFAM46C plasmid as template was cloned into HIVSFFV vector using SLIC approach. To obtain HIVSFFV FLAG-FAM46C construct, PCR product with primers 15_125F1 and HivN_R using pCDNA FAM46CGFP plasmid as template, then reamplified with HivNFlgF and HivN_R and finally was cloned into HIVSFFV vector using SLIC. To make HIVSFFV-RL and HIVSFFV-FL constructs, PCR products obtained with primers 15_131nF/15_131R1 and 15_133_F/15_133_R using pRL-TK (Promega) and pGL3-Control (Promega) templates were cloned into HIVSFFV vector using SLIC approach. To obtain HIVSFFV-RL5Box construct, two PCR product with primers 15_131nF/15_131R1 and 15_131F2/15_131_R using pRL5Box (from Prof. Filipowicz) plasmid as template were cloned in one reaction into HIVSFFV vector using SLIC approach. The HIVSFFV FAM46C-FLAG constructs, was obtained by cloning of PCR product with primers HIV_for and 15_127R1 using pcDNA FAM46CGFP plasmid as template, then reamlified with HIV_for and HivCflgR and finally was cloned into HIVSFFV vector using SLIC approach. To make HIVSFFV GFP-FAM46C construct, GFP sequence was amplified with primers HivNgfpF and 15_127R1, FAM46C was amplified with primers 9_114_F and HivN_R using pCDNA FAM46CGFP plasmid as template for both reactions and finally was cloned into HIVSFFV vector in the same reaction using SLIC approach. All HIVSFFV construct were sequenced from HIV_f, HIV_r. Lentiviral packaging (pMD2.G) and envelope (pPAX2) plasmids were kindly provided by Prof. Didier Trono (École Polytechnique Fédérale de Lausanne, Switzerland). The FAM46CWT, FAM46DWT, and FAM46Cmut constructs for recombinant protein production in Escherichia coli BL21-CodonPlus-RIL strain (Stratagene) were prepared by PCR amplification from previous pcDNA constructs using appropriate primers. The PCR products were subsequently cloned into a modified pET28 vector using the SLIC method, resulting in N-terminal HIStag-SUMOtag fusion.

**Tethering assays**. Tethering assays were performed as previously described[19]. In brief, a day before transfection, 0.75 ml of HEK293 cells were seeded on 6-well plates to achieve about 70–80% confluence on the day of transfection. Next, cells were co-transfected with 100 ng of constructs expressing reporter *Renilla* luciferase (RL-5BoxB or RL), 100 ng of control firefly luciferase (FL, pGL3 plasmid) and 2 µg of plasmid encoding tethered NHA-protein using 5 µl of Lipofectamine 2000 and OPTI-MEM media (Invitrogen) according to manufacturer's instructions. All transfections were repeated at least three times. Twenty-four hours after transfection, cells were collected, lysed and protein levels were analyzed by western blot.

If transfected cells were used solely for the analyses of RNA they were co-transfected only with 0.1 µg pRL-5Box plasmid carrying *Renilla* luciferase (RL) and 2 µg of plasmid encoding tethered NHA-protein cells and then were collected 24 h after transfection. This approach was used to avoid *trans* effects between promoters on co-transfected plasmids which may affect RL reporter gene expression[47]. For special purposes, such as higher yield of RNA, cell cultures and transfections were scaled-up.

**Tethering assay in MM cells**. SKMM1 and H929 cells were co-transduced with lentiviral particles carrying FL and RL or RL5Box and FL at MOI 1. Expression of reporter genes in obtained stable cell lines was verified by western blot then cells were transduced with lentiviruses allowing expression of wild-type and mutated NHA-tagged FAM46C protein at MOI 5. Twenty-four hours after transduction, cells were collected and protein levels was determined by western blot.

**Gene silencing by siRNA and shRNAs.** siRNA-mediated knock-downs were performed using validated Stealth RNAi and Lipofectamine RNAiMAX (both Invitrogen) as previously described[48]. For silencing of BCCIP and PABPC1, three different siRNAs were tested; they are listed in Supplementary Table 3. For further analyses, cells were collected or re-plated 72 h after transfection. The shRNA against human *FAM46C* were purchased from Sigma Aldrich and are listed in Supplementary Table 4. Lentiviral particles were obtained and triter as described below in "Transduction of target cells" section. Cells were transduced at MOI 1 and the third day after transduction were subjected to selection with puromycin (Invitrogen) at 0.5 μg/ml (for H929) and 1 μg/ml (for SKMM1 and RPMI8226) for 7 days to obtain stable cell lines. The level of remaining transcript was determinate by RT-qPCR.

**B cells and MM cells proliferation assay.** Naive B cells have been isolated from 3 polled spleens of sex-matched 12-weeks-old littermate mice using CD43 microbead (Miltenyi Biotec) negative selection and were cultured in RPMI 1640 containing 15% FBS. B cells were activated by the addition of 20 μg/ml LPS (Sigma) and IL-4 (R&D Systems) at 20 ng/ml to the culture media. Both primary murine B-cells and human MM cell lines have been stained with 5 μM CFSE according to manufacturer's instructions and cultured for 72 h or 48 and 72 h, respectively. To calculate the rate of CFSE fluorescent signal dilution autofluorescence, defined as the signal from unstained samples, have been subtracted from the fluorescence of the stained cells and initial (0 h) values have been divided by ones measured at 48 and 72 h. The results were presented as relative to the control transfected with an empty vector.

**FAM46C localization.** The 293T, RAJI, HL60, H929, SKMM1, RPMI8226 cells were transduced with lentiviruses caring FAM46C$^{WT}$ at MOI 1. Forty-eight hours after transduction cells were stained with Hoechst 33342 and subsequently processed. Non-transduced cells were used as auto fluorescence controls A day prior to transfection, cells were seeded to Nunc™ Lab-Tek™ II Chamber Slide™ Systems in order to reach 50% of confluence on the day of transfection. DNA was stained by adding the dye Hoechst 33342 (Invitrogen) to the medium to a final concentration of 2 mg/ml. After 20 min of incubation at 37 °C, the medium was exchanged and cells were prepared for observation. For the intravital observation of the cells, a confocal system (Fluoview FV1000) equipped with a spectral detector (Olympus) and a 60×/1.2 water immersion objective was used. Images were processed using ImageJ software.

**RNA isolation and poly(A) fractionation.** RNA isolation was performed with TRIzol reagent (Invitrogen, 15596) according to the manufacturer's instructions. Poly(A) fractionation was performed as previously described with the following modifications[49]. In brief, 80 μg of total RNA was mixed with 400 μl GTC buffer (4 M guanidine thiocyanate, 25 mM sodium citrate, pH 7.1, 2% β-mercaptoethanol) and 750 pmol of 5′-biotinylated TEG-oligo(dT)$_{25}$ (Future Synthesis). 850 μl of "dilution buffer" (6 × SSC, 10 mM Tris-HCl, pH 7.4, 1 mM EDTA, 0.25% SDS, 1% β-mercaptoethanol) was added, the mixture was incubated at 70 °C for 5 min, and then it was centrifuged at 12,000 rpm for 10 min at RT. The resulting supernatant was mixed with 100 μl of M280 beads (Invitrogen; 60210) washed 3 times with 0.5 × SSC buffer. After binding, samples were washed 3 times with 0.5 × SSC buffer for 10 min. RNA fractions were eluted with decreasing concentrations of SSC buffer (from 0.2 × SSC to 0.025 × SSC) and the final elution was performed with water. RNA was precipitated and subjected to further analysis.

**RNAse H assay.** 20 μg of total RNA was mixed with 100 ng of oligo(dT)$_{25}$ (Invitrogen), heated for 1 min at 95 °C, and transferred to ice. Then, 10 μl of 5× RH buffer (100 mM, 0.5 M KCl, 50 mM MgCl$_2$, 50 mM DTT, and 25% sucrose) was added to 5 U of RNaseH (NEB). Reactions were carried out for 45 min at 37 °C and RNA was recovered by extraction with phenol/chloroform, precipitated, and analyzed using northern blots.

**RNA-seq libraries.** 1 μg of total RNA was treated with Turbo-DNase (Life Technologies) according to the manufacturer's instructions. Then, rRNA was removed from samples using a Ribo-Zero Kit (Epicentre), according to the manufacturer's protocol, and samples were spiked-in with external RNA (ERCC RNA Spike-In Mix; Life Technologies). For libraries prepared from poly(A)+ fractions, the ribo-depletion step was omitted. Finally, RNA libraries were prepared using a TruSeq RNA Sample Preparation Kit (Illumina) or a KAPA Stranded RNA-Seq Library Preparation Kit (Kapa Biosystems) according to the manufacturer's instructions. Quality of libraries was determined by chip electrophoresis performed using the Agilent 2100 Bioanalyzer (Agilent Technologies, Inc.).

**Transduction of target cells.** For lentivirus production, $2.6 \times 10^6$ HEK-293T cells were seeded into 10 cm dishes 24 h prior to transfection in 10 ml of DMEM supplemented with 10% FBS (Invitrogen) and penicillin/streptomycin solution (Sigma). The lentiviruses encoding the gene of interest (GOI) were introduced to the packaging cells by calcium phosphate co-transfection with 8.6 μg of a GOI-containing vector HIV-SFFV and components of 2nd generation of packaging

vectors, namely, 8.6 μg of psPAX2 packaging vector and 5.5 μg of pMD2.G envelope vector. The plasmids were resuspended in 450 μl of 300 mM CaCl$_2$ and added dropwise to 450 μl of 2 × concentrated HEPES-buffered saline (280 mM NaCl, 20 mM HEPES, 1.5 mM Na$_2$HPO$_4$, 10 mM KCl and 12 mM D-glucose pH 7.2) by vortexing. The precipitate was immediately added to the cell culture medium with gentle swirling. The medium was replaced with 6 ml of fresh DMEM 16 h post transduction.

Virus-containing supernatants were collected 24 h later, centrifuged for 3 min at $300 \times g$, and filtered through 0.45 μm low-protein-binding filters (Millipore) Then, the supernatants were concentrated $10 \times$ by centrifugation ($3000 \times g$, 16 h, 4 °C). The titer of viral particles was estimated using a Lenti-X p24 Rapid Titer Kit (Clontech) according to the manufacturer's protocol. Multiplication of infection (MOI) was calculated by dividing the vector titer for the number of cells transduced. Efficiency of transduction was determined by flow cytometry for GFP-expressing cells using an Accuri C6 benchtop cytometer (BD Biosciences). Additional staining with 2.5 μg/ml propidium iodide (Sigma-Aldrich) in PBS was performed to distinguish dead cells. If needed, to select GOI-expressing cells, sorting for GFP-positive cells using the Aria III cell sorter (BD Biosciences) was performed.

**RNA-seq data analysis.** Ribo-depleted total RNA isolated from mutant and wild type H929 cell line quadruplicates were used to prepare strand-specific libraries (dUTP RNA); Ribo-depleted total RNA isolated from mutant and wild type SKMM1 cell line triplicates were also used to prepare non-strand-specific TRUseq RNA libraries. These libraries were subsequently sequenced using an Illumina HiSeq sequencing platform to the average number of 18.5 million reads per sample in 100-nt pair-end mode. Poly(A) tail length fractionation libraries were sequenced to an average depth of in 75-nt pair-end fashion.

Reads were mapped to the reference human genome (hg38) using STAR short-read aligner with default settings (version STAR_2.4.0) yielding an average of 80.2% uniquely mapped reads[50]. The quality control, read processing and filtering, visualization of the results, and counting of reads for the Genecode v22 comprehensive annotation were performed using custom scripts utilizing elements of the RSeQC, BEDtools and SAMtools packages[51, 52]. Differential expression analyses were performed using the DESeq2 Bioconductor R package[53]. For the poly (A) fractionation data, a "polyadenylation ratio" was calculated by dividing the levels of individual mRNA in long poly(A) fractions (#5 and #6) relative to the short fraction (#1) in WT FAM46C cells by equivalent relative levels observed in FAM46C$^{mut}$-GFP cells. Functional clustering of GO terms analysis was performed using DAVID[22]. The search for common sequence motif was performed using HOMER[54] and further analyzed by STAMP[55].

**Multiple alignment.** Multiple alignments analyses were performed using PRofile ALIgNEment (PRALINE) (http://ibivu.cs.vu.nl/programs/pralinewww/)[56, 57].

**Mass spectrometry analysis.** MS analysis were performed essentially as described[58]. Briefly, precipitated proteins were dissolved in 100 μl of 100 mM ammonium bicarbonate buffer, reduced in 100 mM DTT for 30 min at 57 °C, alkylated in 55 mM iodoacetamide for 40 min at RT in the dark, and digested overnight with 10 ng/ml trypsin (Promega) at 37 °C. Finally, trifluoroacetic acid was added at a final concentration of 0.1%. MS analysis was performed by LC-MS in the Laboratory of Mass Spectrometry (IBB PAS, Warsaw) using a nanoAcquity UPLC system (Waters) coupled to an LTQ-Orbitrap Velos mass spectrometer (Thermo Scientific). Peptides were separated by a 180-min linear gradient of 95% solution A (0.1% formic acid in water) to 35% solution B (acetonitrile and 0.1% formic acid). The measurement of each sample was preceded by three washing runs to avoid cross-contamination; the final MS washing run was searched for the presence of cross-contamination between samples. If the protein of interest was identified in the washing run and in the next measured sample at the same or smaller intensity, then the sample was regard as contaminated and these samples were excluded from final graphs. The mass spectrometer was operated in the data-dependent MS-MS2 mode and data were acquired in the $m/z$ range of 300–2000. Data were analyzed with the Max-Quant (Version 1.5.3.12) platform. The reference human proteome database from UniProt was used. We also searched our data for protein isoforms (for example BCCIP alfa/beta) to determine which were present. Label-Free-Quantification (LFQ) intensity values were calculated using the MaxLFQ algorithm to estimate quantities of identified proteins and identified proteins were analyzed as follows. Protein abundance was defined as the LFQ value calculated by MaxQuant software for a protein (sum of intensities of identified peptides of a given protein) divided by its molecular weight. Specificity was defined as the ratio of the protein LFQ intensity measured in the bait purification to background level (i.e., protein LFQ intensity in the negative control purification with the background level was arbitrarily set to 1 for proteins not detected in the negative control). High values of both protein abundance and specificity indicated proteins that were enriched in the purification and thus suggested interactions. Common protein contaminations were removed from graphs. All MS analyses were performed at the Mass Spectrometry Laboratory, IBB PAS.

**Co-Immunoprecipitation**. Cell collected from two 145 mm plates at a confluency of 80–90% were fleshly frozen, thawed on ice, and incubated for 30 min at 4 °C with gentle rotation in 3 ml LB buffer (20mM Tris, 150 mM NaCl, 0.5% Triton-X100, 1 mM DTT, 1 mM PMSF, 0.02 μM pepstatin A, 0.02 μg/ml chymostatin, 0.006 μM leupeptin, and 20 μM benzamidine hydrochloride supplemented with proteases and phosphatase inhibitors; Invitrogen). Next, lysates were sonicated for 30 min in Bioruptor Plus (Diagenode) (30s pulses with 30s intervals), after which they were cleared by centrifugation at 13,000 rpm and 4 °C for 15 min.

Immunoprecipitations were performed using house-in made affinity slurry CNBr-activated SepFast MAG 4HF [Biotoolomics] coupled with house-in purified anti-GFP antibodies. After 2h of incubation, beads were washed six times with LB buffer and finaly washed with TEV buffer (10 mM Tris-HCl pH 7.4, 75 mM NaCl, and 1 mM DTT). Cleavage with TEV protease was conducted for 2 h at RT. Then, proteins were precipitated with PRM reagents (0.05 mM pyrogallol red, 0.16 mM sodium molybdate, 1 mM sodium oxalate, 50 mM succinic acid, and pH 2.5 (Sigma-Aldrich)) and then analyzed by MS.

**Polyadenylation assay**. 150 ng of purified recombinant protein was mixed with $^{32}$P-labeled RNA substrate (A)$_{15}$ in the presence of 1 mM ATP (NEB), 2 U/μl RNasine (Fermentas), 0.5 mM $Mg^{2+}$, or $Mn^{2+}$ (or both) in PAP buffer (25 mM Tris-HCl pH 7.0, 50 mM KCl, 0.02 mM EDTA, 0.2 mM DTT, 100 μg/ml BSA [Invitrogen], and 10% glycerol). Reactions were carried out at 37 °C and stopped at indicated time points by addition of equal volume of RNA loading dye (98% deionized formamide, 25 mM EDTA pH 8.0, 0.025% (w/v) xylene cyanol, and 0.025% (w/v) bromophenol blue). Reaction products were separated in 8 M urea/15% PAGE in 0.5× TBE.

**RNA-protein UV-crosslinking and immunoprecipitation**. FAM46C$^{WT}$GFP crosslinking to RNA followed by immunoprecipitation was performed as described[59]. Briefly, five 145 mm plates of stable HEK293 Flp-In TREx cell line expressing FAM46C$^{WT}$GFP at 90% confluency were irradiated with 254 nm UV light (2500 μj/cm$^2$; UVP Crosslinker). Protein cross-linked to RNA were then immunoprecipitated with anti-GFP resin, followed by RNA labeling with $^{32}$P, SDS-PAGE complexes separation, transferred to the nitrocellulose membrane, staining with Ponceau S, and autoradiographed.

**Mice generation**. Basing on UCSC Mice GRCm38/mm10 Assembly, sgRNA (chimeric single-guide RNA) were designed as close as possible to the catalytic site of FAM46C gene to generate knock-out and to the STOP codon to generate FLAG-tag knock-in mice. According to the MIT CRISPR online design tool (crispr.mit. edu) our guides, located chr3:100473168-100473187 and chr3:100472250-100472273, reached score of 82 and 57 points, respectively, with very low risk of off-targets (for guide RNA sequences see Supplementary Data 4). For sgRNA T7 RNA Polymerase based synthesis we used mFam46C_mut_sgRNA_f primer for FAM46C knock-out line or mFam46C_FLAG_sgRNA_f primer for FAM46C-FLAG knock-in line and Universal_sgRNA_rev primers (underlined is T7 RNA polymerase promoter).

**Cas9 mRNA synthesis**. *Streptococcus pyogenes* Cas9 CDS was amplified from PX458 (pSpCas9(BB)-2A-GFP, Addgene Plasmid #48138) using T7_SpCas9_for and SpCas9_rev primers and subsequently transcribed with T7 RNA polymerase.

**Collection of zygotes**. Ethical approval for the procedures on animals was obtained from I Local Ethical Commission for Experiments on Animals in Warsaw (decisions no. 527/2013 and 176/2016). Embryos used in all experiments were isolated from F1(C57BL/6×CBA) mouse females, which were induced to super-ovulate by injection of 10 IU of PMSG (Pregnant Mare Serum Gonadotropin; Folligon, Intervet, Netherlands) and 10 IU of hCG (Human Chorionic Gonadotropin; Chorulon, Intervet, Netherlands) 48–52 h later. Females were mated with males of the same strain immediately after hCG injection. Zygotes were collected from mated females 21–22 h post hCG injection. They were released from the oviducts into the hyaluronidase solution (300 μg/ml, Sigma) to disperse follicular cells. Follicular cells-free zygotes were washed and incubated in drops of M2 medium, under mineral oil (Sigma), at 37.5 °C, 5% CO$_2$ in the air until being used for microinjections[60].

**Cocktail preparation and zygote injection**. Injection mixes were prepared on ice prior to use and were never frozen and used again. Microinjections were performed on zygotes 24–25 h post hCG, in a drop of M2 medium under mineral oil. Zygotes, were microinjected into the cytoplasm using Eppendorf 5242 microinjector (Eppendorf-Netheler-Hinz GmbH) and Eppendorf Femtotips II capillaries with the following CRISPR cocktail: Cas9 mRNA 25 ng/μl, sgRNA 15 ng/μl, and mFam46C_FLAG_oligo. Injected embryos were incubated in M2, in standard culture conditions for 30 min and next transferred into drops of M16 medium (Sigma) and cultured overnight.

**Embryo transfer**. After overnight culture microinjected embryos at 2-cell stage were transferred into the oviducts of 0.5-day p.c. pseudo-pregnant females. Usually, embryos were transferred into one oviduct only, and no more than 10–12 embryos into one female. Foster mothers were kept in individual cages until delivery of the pups. Four-weeks-old young were separated from foster mothers. Small fragments of their ears or tails were taken from anaesthetized animals to perform genotyping. Males and females from one foster mother were kept in separate cages.

**Mice genotyping**. Genomic DNA from ear or tail fragments was isolated and PCR was performed with Fam46C_seqF and Fam46C_seqR primer pair for detecting deletions of the catalytic domain and mF46cFLAG_seq1F, mF46cFLAG_seq1R primer pair for genotyping FLAG-tag insertions, DNA sequencing results were analyzed with Mutation Surveyor 4.0 software Demo (SoftGenetics). FAM46C mice genotypes were shown in Supplementary Table 5.

**Bone marrow analyses**. To determine bone marrow cellularity, the cells were counted on a Bürker hemocytometer and dead cells were excluded using trypan blue dye. To evaluate percentage of plasma cells bone marrow cells were blocked in 2% BSA for 15 min. Next, the cells were labeled with anti-CD45.2 Horizon V500 (BD 562129) and anti-CD138-PE (BD 553714) monoclonal antibodies for 30 min at room temperature at 1:25 dilution, then washed with PBS and analyzed using LSR Fortessa and BD FACSDiva software version 8.0.1. The plasma cells were identified using CD138 from CD45.2-positive population. Discriminating positive vs. negative signals was ensured with fluorescence minus one (FMO) control.

**Blood analyses**. The blood samples were collected from 12-weeks-old WT and *Fam46C* KO mice of both sexes and were commercially analyzed at Veterinary Diagnostic Laboratory LabWet in Warsaw (http://www.labwet.pl/) on the day of blood collection. Statistical analysis of the results was performed using GraphPad Prism6 software.

**Splenocytes fractionation**. Cell fractionation was performed using a REAP protocol[61]. Briefly, cells isolated from spleen of WT and FLAG-tagged knock-in mice were washed in ice-cold PBS then resuspended in ice-cold 0.1% NP40-PBS (Sigma) and centrifuged in benchtop microfuge at maximum speed for 10 s. Supernatant was collected as "cytosolic fraction" and remaining pellet was subsequently washed once with 0.1% NP40-PBS. Next nuclei pellet was resuspended in 0.1% NP40-0.5×PBS containing 500 U of viscolase (A&A Biotechnology), incubated at 37 °C for 30 min and designated as "nuclear fraction". Collected fractions were supplemented with Laemmli sample buffer, separated in SDS-PAGE gels and analyzed with western blot approach.

**Data availability**. The RNA-seq data from this study have been submitted to the NCBI Gene Expression Omnibus (GEO; http://www.ncbi.nlm.nih.gov/geo/query/acc.cgi?acc=GSE83772) under accession number GSE83772. The mass spectrometry proteomics data have been deposited to the ProteomeXchange Consortium via the PRIDE partner repository with the dataset identifier PXD006790. All scripts used in bioinformatics analyses are available for researchers upon request. All other remaining data are available within the Article and Supplementary Files or available from the authors upon request.

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

## Acknowledgements

We thank Els Verhoeyen for the HIVSFFVRFP plasmid, Michael Kuehl for the SKMM1 cells, Witold Filipowicz for the pClneo-NHA and reporter plasmids, Didier Trono for the pMD2.G and pPAX2 plasmids, Kasia Kowalska for help with DNA cloning, Krystian Bijata for recombinant protein purification, Aleksander Chlebowski for microscopy assistance and reading of the manuscript, and AD laboratory members for stimulating discussions. The equipment from www.ibb.waw.pl/en/services/mass-spectrometry-lab was sponsored in part by the Centre for Preclinical Research and Technology (CePT), a project co-sponsored by European Regional Development Fund and Innovative Economy, The National Cohesion Strategy of Poland. This work was funded by ERC Starting Grant 309419 PAPs & PUPs; the FNP fellowships 'Ideas for Poland' and 'Master scholarships' (to A.D.), scholarship for outstanding young scientists from the Polish Ministry of Science and Higher Education and "Iuventus Plus" grant IP2012 046272 (0462/IP1/2013/72) (to S.M.). Experiments were carried out with the use of CePT infrastructure financed by the European Union the European Regional Development Fund [Innovative economy 2007-13, Agreement POIG.02.02. 00-14-024/ 08-00].

## Author contributions

A.D.: Developed and directed the studies, S.M.: Performed the vast majority of the biochemical and cell lines experiments, J.C. with help of S.M. transduced MM cells, J.C. performed flow cytometry analyses, T.M.K.: Performed all bioinformatics analyses and proliferation assays, O.G.: Analyzed mice phenotypes and participated in localization studies, D.C.: Performed all high resolution mass spectrometry analyses, V.L.: Made preliminary RNA-seq libraries, J.G.: Designed and prepared CRISPR reagents, genotyped MM cells and mice, and E.B. generated transgenic animals. A.D. and S.M.: Wrote the manuscript with minor contributions from all authors. D.N.: Supervised J.C.'s work and discussed results.

## Additional information

**Competing interests:** The authors declare no competing financial interests.

