## [Peer Review file · Nature Communications]

Reviewers' comments:

Reviewer #1 (Remarks to the Author):

The authors present a variety of strands of evidence suggesting that the FAM46C gene, mutations in which were previously suggested to be involved in multiple myeloma, encodes a non-canonical and possibly cytoplasmic poly(A) polymerase. They further suggest on the basis of their data that poly(A) tail shortening of a sub-set of mRNAs, as a result of FAM46C mutation, may contribute directly to the development of multiple myeloma. The novelty and general interest of this report rests partly on the identification of a novel sub-class of poly(A) polymerases, and partly on the notion that altered polyadenylation may contribute directly to tumorigenesis.

The wide-ranging study addresses numerous aspects of FAM46C function. In general, the aspects dealing with poly(A) tail length changes and mRNA abundance in cells are more robust and convincing than the characterisation of FAM46C/D activity *in vitro*, investigation of FAM46C subcellular localisation and the experiments addressing the effects of FAM46C mutation on cell proliferation and death. One general limitation is that many of the cell-based experiments involve expression of exogenous FAM46C in a wild-type or mutant form without any indication of the level of expression of these exogenous forms relative to what would normally be found in the cell lines used. The authors should in addition be encouraged to address the following points:

1. Fig. 1a, b. The amount of polyadenylated product does not seem to increase with increased incubation time. Though it is quite hard to assess accurately, most of the products seem to have between one and four adenosine residues added even after protracted incubation with FAM46C or FAM46D. Is this modest increase in poly(A) tail length likely to be biologically significant, and does it suggest that the activity of these enzymes is non-processive?
2. Fig. 1d. Localization of FAM46C appears primarily nuclear. On what basis is mixed nuclear and cytoplasmic localization "expected for non-canonical poly(A) polymerases" (line 121)? At the very least, the statement in the abstract that FAM46C encodes a cytoplasmic enzyme should be qualified to reflect the data.
3. Figs 2 and (especially) 3 will be difficult for the general readership to interpret as they stand. For example, the text indicates that "greater [levels of] RL mRNA molecules were present in fractions with longer poly(A) tails" (line 135), but how the data presented in Fig. 2d support this interpretation is not explained. Fig. 3a is described in the legend as an "example gating strategy" without further explanation. I could guess that FSC is forward scatter, SSC side scatter and that PI-A is a measure of red fluorescence, with exclusion of PI being used as a measure of cell viability, but all these points should be properly explained.
4. While the data presented in Fig. 3 are suggestive, it seems premature to conclude that "FAM46C acts as a growth suppressor specific to MM cells" (line 185). Was the level of endogenous or ectopic expression of FAM46C equivalent in all the cell lines tested, and what contributions might the numerous genetic differences between the cell lines tested (as opposed to the tumor type from which they were derived) make to their response to FAM46C expression?
5. Fig. 6b. As anti-GFP antibody was used in the immunoprecipitations, it is not clear why no GFP signal is visible in the western blot of the 'GFP only' control. Has it been cut off, or is it absent because it lacks a TEV cleavage site? In its absence, the specificity of the BCCIP and PABPC1 signals in the FAM46C lanes is less thoroughly established.
6. There are a number of typographic and grammatical errors throughout the manuscript, which should be thoroughly revised in case the intended meaning is not clear. As examples, 'CpU-reach' (line 238) should read 'CpU-rich'; 'papstain' and 'laupeptin' (line 592) should presumably read 'pepstatin' and 'leupeptin'.

Reviewer #2 (Remarks to the Author):

Mroczek et al. report on "The FAM46C gene encodes a non-canonical poly(A) polymerase and acts as an oncosuppressor in multiple myeloma" Next generation sequencing in primary multiple myeloma (MM) cells has recently detected mutations in the FAM46C gene with a pattern most consistent with tumor suppressor function. The function of FAM46C was unknown at the time of these reports. The authors here focus their work in demonstrating that the FAM46C gene is a non-canonical poly(A) polymerase and attempt identifying the downstream events responsible for its tumor suppressor function in MM.

Major concerns:

1. The authors do not acknowledge and appear not to be familiar with three published scientific works on FAM46C. FAM46C was cloned and characterized in its function as an mRNA-stabilization protein by Meng Tian, a graduate student in the Biological and Biomedical Sciences PhD program at Harvard. In her published dissertation thesis, Tian demonstrated that the FAM46C gene is necessary for terminal differentiation of erythrocytes by interacting with PABPC1 to increase poly(A) and stabilize hemoglobin mRNA. In their recently published paper titled "FAM46C proteins are novel eukaryotic non-canonical poly(A) polymerases", Kuchta et al. classify FAM46C as a non canonical polyA polymerase based on bio-informatic tools and predict targets and functional consequences of FAM46C activation. Finally, the group of Dr. Keith Stewart presented an oral abstract on the role of FAM46C as a tumor suppressor gene in MM at the last American Society of Hematology (ASH) annual meeting. Such abstract is published in Blood.

In the context of these published studies, the downstream targets of FAM46C in myeloma responsible for its tumor suppressor function have not been clearly identified. As such, there was an opportunity for the authors to provide a major contribution to the field by dissecting the interacting partners and downstream element of FAM46C in myeloma. Unfortunately, the authors were not able to clearly demonstrate the molecular targets of FAM46C in MM, thus making minimal contribution to the field.

2. HEK293 was used by the authors to validate the function of FAM46C (figure 1 and 2) and its interacting partners (figure 6b and supplementary figures 5 and 6). Based on published data, it is reasonable to expect that interacting partners for FAM46C may vary in different tissues and that polyadenylation of certain targets may have a tissue-specific effect. For this reason, FAM46C interacting partners identified via IP and MS in SKMM1 and H929 should have been validated via co-IP and western in these same cell lines. Similarly, tethering experiments of FAM46C and its interacting partners PABPC1 and BCCIP β should have been conducted in MM cell lines.

- 3- Based on the data presented, I believe that authors do not provide sufficient evidence that FAM46C is a tumor suppressor gene in MM. While they show that its overexpression is cytotoxic, they do not provide any evidence that its knock out/down is carcinogenic or promotes survival/proliferation. In fact, the need for a FAM46C KO animal model to study its biology should at least be discussed.

Minor points:

1. The authors provide no data regarding the protein expression level of FAM46C in MM cell lines or primary MM cells. Western blots or IHC staining would have been the starting point to support the statement that FAM46C is a tumor suppressor in MM. Furthermore, if one or more FAM46C high-expressing cell lines were identified, KD studies or Crispr-Cas9 mediated KO of FAM46C would have further supported the authors hypothesis.

2. Similarly there is no data regarding FAM46C expression level and/or presence of mutations in HEK293T.

3. Do the authors have knowledge regarding whether FAM46C mutations present in H929 and SKMM1 give rise to a truncated, but functional protein or whether mRNA undergoes decay and never translates? This is an important point as it is very possible that a mutant, but functional FAM46C may act as a dominant negative and impair the function of overexpressed WT FAM46C.

The dominant negative effect hypothesis appears supported by figure 2e where decay of RL probe appears accelerated in HEK293 transfected with mutant FAM46CD90A, D92A compared to mock-transfected cells.

-4. Figure 3, panel A and B: the GFP histograms in panel B are confusing. Considering the high lethality of FAM46C overexpression, the high % of GFP is likely to represent autofluorescent, early apoptotic cells. The authors to GFP sort cells 24-48 hours post transfection and only perform DAPI/Annexin V-APC staining for apoptosis in this sorted population.

5. Figure 3, panel C: given that HL60 and Raji have a low level of expression of the transgene, the authors can not really comment on the lack of effect of forced FAM46C expression on these cell lines. Again, early sorting of GFP-transfected cells and longitudinal follow up for apoptosis would be a better-designed experiment.

6. Supplementary figure 2, panel A: why is there such an early toxicity in POLS and Gld2 transfected cells? Percentage of PI negative is very different at day 3 compared to PAPOLA despite similar levels of GFP+. Please comment.

7. Loading controls (GAPDH, actin..) should be shown in western blots.

8. The complete results of RNAseq and proteomic studies should be made available to readers as supplementary materials.

9. The authors use TEV-eGFP-tagged expression vectors throughout the paper. To exclude interference of the rather large eGFP tag, a C-terminus and N-terminus variant of the tag should be used and consideration should be given to repeat experiments with a 3xFLAG or V5-tagged vector.

10. Statements on page 3, line 59 "a gene whose function is unknown"; page 3, line 74 and line 77-78 and 80-81 should be modified to acknowledge information provided by the above mentioned published material.

11. Statement on page 4, line 90 is inaccurate. Please see Kuchta et al, and modify accordingly.

Reviewer #3 (Remarks to the Author):

In this current study, Mroczek et al. identify FAM46C, a gene encoding a novel poly(A) polymerase and investigate its binding partners, substrates and potential role in multiple myeloma. They show that FAM46C poly(A) polymerase activity is important for the stability of its targets and is dependent on its catalytic activity. They provide some evidence proposing that the enzymatic activity of this protein is important for the cell survival in different MM cell lines. Additionally, they also identify the RNA targets of FAM46C and validate some of them. Despite the fact that they could not provide any hints of a molecular mechanism accountable for the specificity of FAM46C to target a subset of mRNAs, they characterize a novel poly(A) polymerase and highlight its importance for cell survival. In conclusion, although the authors have original concepts, some of the experiments lack significant controls. Major revision should be in place before this manuscript can be considered for publication.

Major Concerns:

In Figure 1a, the authors perform an in vitro polyadenylation assay with purified WT and mutant FAM46C protein. They should also provide the gel showing equal expression of purified proteins, in order to conclude that the catalytic mutant is deficient in polyadenylation. As they discussed in the text, the activity of FAM46C is very low. Did the authors try to change the purification methods or try to purify the protein from cells by pull down and repeat the polyadenylation assay with the cell extract? They are using FAM46D as a control, however they do not give any information about this protein. Are they similar? Is this gene also mutated in MM? This will be important to address in the manuscript.

In Figure 2, the authors used the tethering assay, which is widely accepted as a powerful tool to investigate the role of a protein of interest in mRNA stability. However, the authors did not use

proper controls for the assay. As R-Luc reporter is transfected, another reporter, for example F-Luc, should be used as an internal control, and the RNA and protein levels (generally measuring the luciferase activity) should be normalized accordingly. The internally transfected reporter should also be used for the control in Northern blots, instead of methylene blue staining. Additionally, an R-Luc reporter that does not contain any Box B element should also be used as negative control, as some LN tagged protein has intrinsic activity on this reporter. Moreover, the authors should also provide the quantification of at least 3 independent experiments for Northern blots and show the standard deviations. In Figure 2f, the half-life of the reporter should also be calculated.

In Figure 3, the authors stably overexpress GFP tagged version of WT and mutant FAM46C. They should provide a western blot showing the relative expression of overexpressed protein compared to the endogenous one. This is especially important for Raji and HL60 cells, to be sure that the proteins are actually expressing. In Figure 3a, according to the text, the authors overexpressed GFP alone as a control, however the cells do not express GFP (GFP+: 0.3%), or perhaps there is labeling problem. In the same figure, the intensity of GFP-WT protein is very low, so it is crucial to check the expression of the protein at different stages. It is possible to have weak expression because the cells are dying, however the expression should be checked in an early point. Otherwise, the experiment is not conclusive.

To verify the role of FAM46C as an onco-suppressor, as mentioned in the title, did the authors perform any knock-down assays of FAM46C to test whether this contributed to cellular transformation?

In Figure 5, the authors perform a pull down to identify interacting partners of FAM46C. Did they perform this IP in the presence of RNases? If not, how are they sure that the interaction is not mediated via RNA? More importantly, how do they explain that PABP is unidentifiable in their MS, despite the high abundance of the protein. If this is because of the IP condition, did they try to change the condition accordingly? Do they validate the interaction with the other proteins?

Minor Concerns:

In the introduction, the authors state: "The entire sequence is altered by mutations except for the short 5' coding." What does this mean? Do they mean that the 5' region of the CDS is not altered or that the 5' UTR is not altered? They should describe this more clearly.

The authors should also summarize their results and the novelty of their findings properly in the introduction.

In Figure 1d, the authors declare that FAM46C is expressed in the nucleus and cytoplasm and mention that the protein may have RNA stabilizing and destabilizing functions. However, they provide very poor quality confocal images. These images should be replaced, and control cells (without GFP) should also be included.

In Figure 2e, the control does not reflect the quantification.

Lane 150: Is it MM1.S or RPMI8226?

In Fig 5b, the asterisk should be explained.

In the text, it would be easier for the reader to follow if the authors described the only mutant they use once, and then just label it as "mut".

Mroczek et al., point by point response to the Reviewer`s comments

Reviewers' comments:

Reviewer #1 (Remarks to the Author): expert in poly(A) polymerase

The authors present a variety of strands of evidence suggesting that the FAM46C gene, mutations in which were previously suggested to be involved in multiple myeloma, encodes a non-canonical and possibly cytoplasmic poly(A) polymerase. They further suggest on the basis of their data that poly(A) tail shortening of a sub-set of mRNAs, as a result of FAM46C mutation, may contribute directly to the development of multiple myeloma. The novelty and general interest of this report rests partly on the identification of a novel sub-class of poly(A) polymerases, and partly on the notion that altered polyadenylation may contribute directly to tumorigenesis.

The wide-ranging study addresses numerous aspects of FAM46C function. In general, the aspects dealing with poly(A) tail length changes and mRNA abundance in cells are more robust and convincing than the characterisation of FAM46C/D activity *in vitro*, investigation of FAM46C subcellular localisation and the experiments addressing the effects of FAM46C mutation on cell proliferation and death. One general limitation is that many of the cell-based experiments involve expression of exogenous FAM46C in a wild-type or mutant form without any indication of the level of expression of these exogenous forms relative to what would normally be found in the cell lines used.

We thank the Reviewer for an overall positive opinion about our paper. Regarding the measurements of the levels of the endogenous protein the reason we did not verify endogenous FAM46C levels in MM cells expressing wild-type FAM46C was that our intensive trials to raise or purchase specific anti-FAM46C antibodies failed. Thus, to study the expression of FAM46C in the B cell lineage, from which MM originate, we have generated a FAM46C FLAG knock-in mouse. The animals did not display any detectable phenotypes. We performed western blot analysis of activated splenocytes, which revealed that FAM46C expression was very strongly induced upon activation. Furthermore, cell fractionation experiments revealed that the protein is localized both in the cytoplasm and nucleus, which corroborates our localization studies by transfection of FAM46C-GFP expressing constructs.

The authors should in addition be encouraged to address the following points:

1. Fig. 1a, b. The amount of polyadenylated product does not seem to increase with increased incubation time. Though it is quite hard to assess accurately, most of the products seem to have between one and four adenosine residues added even after protracted incubation with FAM46C or FAM46D. Is this modest increase in poly(A) tail length likely to be biologically significant, and does it suggest that the activity of these enzymes is non-processive?

FAM46 proteins have indeed relatively poor processivity *in vitro*. Such a statement is now included in the revised version of the manuscript. Based on the analysis of the endogenous substrates (RNA-seq after poly(A) fractionation and Northern blots), we expect that FAM46 proteins are processive *in vivo* due to additional factors that increase FAM46 processivity.

2. Fig. 1d. Localization of FAM46C appears primarily nuclear. On what basis is mixed nuclear and cytoplasmic localization "expected for non-canonical poly(A) polymerases" (line 121)?

We have repeated localization experiments with 6 different cell lines. As can be seen on a new Figure 2 FAM46C has a clear mixed nuclear and cytoplasmic localization. Furthermore, cell fractionation experiment performed on primary splenocytes isolated from FAM46C FLAG knock-in mice revealed that it is localized both in the cytoplasm and nucleus, which corroborates our localization studies based on confocal imaging.

At the very least, the statement in the abstract that FAM46C encodes a cytoplasmic enzyme should be qualified to reflect the data.

We have removed such a statement from the abstract.

3. Figs 2 and (especially) 3 will be difficult for the general readership to interpret as they stand. For example, the text indicates that "greater [levels of] RL mRNA molecules were present in fractions with longer poly(A) tails" (line 135), but how the data presented in Fig. 2d support this interpretation is not explained.

The sentence has been rephrased.

Fig. 3a is described in the legend as an "example gating strategy" without further explanation. I could guess that FSC is forward scatter, SSC side scatter and that PI-A is a measure of red fluorescence, with exclusion of PI being used as a measure of cell viability, but all these points should be properly explained.

The figure legend was rewritten.

4. While the data presented in Fig. 3 are suggestive, it seems premature to conclude that "FAM46C acts as a growth suppressor specific to MM cells" (line 185). Was the level of endogenous or ectopic expression of FAM46C equivalent in all the cell lines tested, and what contributions might the numerous genetic differences between the cell lines tested (as opposed to the tumor type from which they were derived) make to their response to FAM46C expression? Our intensive trials to raise specific anti-FAM46C antibodies failed. Moreover, we have examined all commercially available ones and none of them was sensitive enough. Thus, we were unable to quantify levels of endogenous proteins. Therefore, in order to analyze the endogenous protein we have generated a FAM46C FLAG knock-in mouse (See above).

5. Fig. 6b. As anti-GFP antibody was used in the immunoprecipitations, it is not clear why no GFP signal is visible in the western blot of the 'GFP only' control. Has it been cut off, or is it absent because it lacks a TEV cleavage site? In its absence, the specificity of the BCCIP and PABPC1 signals in the FAM46C lanes is less thoroughly established.

The blot was simply cut off to avoid very strong signal which could influence the readout. Such information is now included in the new Figure 9.

6. There are a number of typographic and grammatical errors throughout the manuscript, which should be thoroughly revised in case the intended meaning is not clear. As examples, 'CpU-reach' (line 238) should read 'CpU-rich'; 'papstain' and 'laupeptin' (line 592) should presumably read 'pepstatin' and 'leupeptin'.

We apologize for all typing errors. The revised version of the manuscripts has been edited.

Reviewer #2 (Remarks to the Author): Expert in multiple myeloma

Mroczek et al. report on "The FAM46C gene encodes a non-canonical poly(A) polymerase and acts as an oncosuppressor in multiple myeloma" Next generation sequencing in primary multiple myeloma (MM) cells has recently detected mutations in the FAM46C gene with a pattern most consistent with tumor suppressor function. The function of FAM46C was unknown at the time of these reports. The authors here focus their work in demonstrating that the FAM46C gene is a non-canonical poly(A) polymerase and attempt identifying the downstream events responsible for its tumor suppressor function in MM.

Major concerns:

1. The authors do not acknowledge and appear not to be familiar with three published scientific works on FAM46C. FAM46C was cloned and characterized in its function as an mRNA-stabilization protein by Meng Tian, a graduate student in the Biological and Biomedical Sciences PhD program at Harvard. In her published dissertation thesis, Tian demonstrated that the FAM46C gene is necessary for terminal differentiation of erythrocytes by interacting with PABPC1 to increase polyA and stabilize hemoglobin mRNA. In their recently published paper titled "FAM46C proteins are novel eukaryotic non-canonical poly(A) polymerases", Kuchta et al. classify FAM46C as a non canonical polyA polymerase based on bio-informatic tools and predict targets and functional consequences of FAM46C activation. Finally, the group of Dr. Keith Stewart presented an oral abstract on the role of FAM46C as a tumor suppressor gene in MM at the last American Society of Hematology (ASH) annual meeting. Such abstract is published in Blood.

The study by Kuchta et al. was cited by us in the original version of the manuscript as well as the previous publication from 2009, which actually first assigned FAM46C as a potential nucleotidyl transferase. However, both papers do not go beyond bioinformatic analysis, which cannot predict if FAM46C is a poly(A) or poly(U) polymerase or even an enzyme catalyzing reactions not related to RNA.

FAM46C was identified in forward genetic screen, where its mutation resulted in anemia. The results are included in the PhD thesis from Mang Tian, defended in 2010, but were never published in a peer-reviewed journal. This is the reason why it was not cited by us. However, it is now included in the references, according to the Reviewer's suggestion. Actually, our newly generated FAM46C KO mouse model recapitulated some of the described phenotypes (see below), suggesting that our previous results were indeed valid.

As for the conference abstract, we leave the decision to the scientific Editor, as we believe that it is not a common practice to cite conference abstracts that were not peer reviewed and do not present supporting data].

In the context of these published studies, the downstream targets of FAM46C in myeloma responsible for its tumor suppressor function have not been clearly identified. As such, there was an opportunity for the authors to provide a major contribution to the field by dissecting the interacting partners and downstream element of FAM46C in myeloma. Unfortunately, the authors were not able to clearly demonstrate the molecular targets of FAM46C in MM, thus making minimal contribution to the field.

We absolutely disagree with such a statement. Our RNA-seq experiments combined with poly(A) tail fractionation allowed us to identify the FAM46C targets in MM cell lines.

2. HEK293 was used by the authors to validate the function of FAM46C (figure 1 and 2) and its interacting partners (figure 6b and supplementary figures 5 and 6). Based on published data, it is reasonable to expect that interacting partners for FAM46C may vary in different tissues and that polyadenylation of certain targets may have a tissue-specific effect. For this reason, FAM46C interacting partners identified via IP and MS in SKMM1 and H929 should have been validated via co-IP and western in these same cell lines. Similarly, tethering experiments of FAM46C and its interacting partners PABPC1 and BCCIP β should have been conducted in MM cell lines. FAM46C interacting partners identified via IP and MS in SKMM1 and H929 were validated via co-IP and western in these same cell lines (these results are now included in the revised version of the manuscript, Figure 9c).

Regarding the tethering experiments, such experiments in MM cell lines are very time consuming because of the need to construct several lentiviral vectors. Since we have not seen very strong effect of tethering of PABPC1 and BCCIP β in HEK293 cells while other experiments in this cell line were recapitulated in MM we expect that they will be valid for MM cell lines.

3. Based on the data presented, I believe that authors do not provide sufficient evidence that FAM46C is a tumor suppressor gene in MM. While they show that its overexpression is cytotoxic, they do not provide any evidence that its knock out/down is carcinogenic or promotes survival/proliferation. In fact, the need for a FAM46C KO animal model to study its biology should at least be discussed.

We have performed shRNA mediated silencing experiments in a RPM8226 MM cell line that expresses wild-type FAM46C. Lowering the FAM46C expression does indeed, as expected, led to increased growth rate of this MM cell line (the data are now included as a Figure 4d,e). Furthermore, we have generated a FAM46C KO animals using CRISPR. Even though detailed analysis of the FAM46C KO phenotypes are outside of the scope of this study, analysis of activated primary lymphocytes revealed that FAM46C mutation lead to an increased proliferation rate (the data are now included as a Figure 6i). Moreover, several hematologic abnormalities were detected (Figure 6 a-h). *FAM46C*^{-/-} mice had significantly lower hemoglobin levels than age- and sex-matching wild-type controls (on average 8.43 mmol/l versus 10.50 mmol/l respectively), suggesting that they suffer from anemia. The red blood cell values were increased in *FAM46C*^{-/-} mice (on average 13.28 T/L in *FAM46C*^{-/-} compared to 10.43 T/L in wild-type mice). MCV, MCH and MCHC were all significantly decreased in *FAM46C*^{-/-} mice (on average 45.75 fL, 1.01 fmol, 21.89 mmol/l respectively in wild-type, compared to 33.42 fL, 0.64 fmol and 18.98 mmol/l respectively in *FAM46C*^{-/-} mice). The microcytic and hypochromic erythrocytes imply that insufficiency of hemoglobin production is the most probable cause of the anemia. The MCV/ RBC ratio (Mentzer index) is higher in wild-type (4.39) than *FAM46C*^{-/-} (2.52), suggesting anemia is a consequence of a block in globin synthesis rather than iron insufficiency, but whether this is true and at what stage the blockage appears would have to be further experimentally tested. The above mentioned results are generally in agreement with the unpublished work from the Fleming laboratory described in Tian PhD thesis. (mentioned by the Reviewer).

Concluding, we are strongly convinced the new data provided by us, allow us to claim that FAM46C is a bona fide oncosuppressor.

Please note: *FAM46C* KO animals generated by us were viable, similarly to the mutant isolated by Fleming. In contrast, according to the Mouse Phenotypic Consortium *FAM46C* is an essential gene. We suppose that the discrepancy may be related of some linked lethal mutation generated by the consortium.

Minor points:

1. The authors provide no data regarding the protein expression level of FAM46C in MM cell lines or primary MM cells. Western blots or IHC staining would have been the starting point to support the statement that FAM46C is a tumor suppressor in MM.

As described above, we were unable to generate or purchase anti-FAM46C antibodies of sufficient quality to measure endogenous expression levels of FAM46C. However, as described above, we have used the newly generated FAM46C FLAG knock-in mouse strain, in which we show that FAM46C is strongly induced during activation of B lymphocytes.

Furthermore, if one or more FAM46C high-expressing cell lines were identified, KD studies or Crispr-Cas9 mediated KO of FAM46C would have further supported the authors hypothesis. See above

2. Similarly there is no data regarding FAM46C expression level and/or presence of mutations in HEK293T.

mRNA for FAM46C is expressed in HEK293 cells (see genome browser snapshot).

Furthermore, there are no mutations in FAM46C in this cell line (<http://hek293genome.org/v2/>)

3. Do the authors have knowledge regarding whether FAM46C mutations present in H929 and SKMM1 give rise to a truncated, but functional protein or whether mRNA undergoes decay and never translates?

Such information was provided as a Supplementary Table in the original version of the manuscript. In both cases there are homozygous frameshift mutations (73 aa in H929 and 191 aa in SKMM1).

This is an important point as it is very possible that a mutant, but functional FAM46C may act as a dominant negative and impair the function of overexpressed WT FAM46C.

Such early truncations cannot produce a functional protein. Furthermore, it is quite unlikely that short fragments broken in the middle of the fold will produce stable entities with dominant negative properties. Importantly, shRNA mediated silencing of FAM46C in cell line with FAM46C frameshift mutations showed small effect on the cell growth, as only one out of three

shRNAs inhibited proliferation indicating that mutations found in these cell lines indeed do not cause dominant negative effects.

The dominant negative effect hypothesis appears supported by figure 2e where decay of RL probe appears accelerated in HEK293 transfected with mutant FAM46CD90A, D92A compared to mock-transfected cells.

In our opinion the difference between mock transfection and transfection with the mutant is marginal and obviously not statistically significant.

4. Figure 3, panel A and B: the GFP histograms in panel B are confusing. Considering the high lethality of FAM46C overexpression, the high % of GFP is likely to represent autofluorescent, early apoptotic cells. The authors to GFP sort cells 24-48 hours post transfection and only perform DAPI/Annexin V-APC staining for

As shown in gating strategy (Fig 4a), GFP fluorescence was analyzed only in PI negative population to reduce influence of autofluorescent apoptotic cells. Description of the Figure has been clarified.

5. Figure 3, panel C: given that HL60 and Raji have a low level of expression of the transgene, the authors can not really comment on the lack of effect of forced FAM46C expression on these cell lines. Again, early sorting of GFP-transfected cells and longitudinal follow up for apoptosis would be a better-designed experiment.

Confocal microscopy analyses confirmed expression of FAM46C in HL60 and Raji cells. Stable cell lines were obtained for both. All these analyses are included in the manuscript (Figure 2). The expression is also confirmed by western blot (see below). However, we cannot exclude the possibility that the lack of toxicity of FAM46C in HL60 and Raji cells is not related to lower expression of the transgene. Such a statement is now included in the revised version of the manuscript.

6. Supplementary figure 2, panel A: why is there such an early toxicity in POLS and Gld2 transfected cells? Percentage of PI negative is very different at day 3 compared to PAPOLA despite similar levels of GFP+. Please comment.

We do not have a perfect explanation for early toxicity of transfections in this particular experiment. However, since after day 3 cells begin to grow normally and stable cell lines can be derived, it is rather related to transduction itself but not overexpression of analyzed poly(A) polymerases. We have softened our statement in the manuscript.

7. Loading controls (GAPDH, actin..) should be shown in western blots.

All appropriate loading controls were included for western and northern blots.

8. The complete results of RNAseq and proteomic studies should be made available to readers as supplementary materials.

Such data were provided in the original submitted manuscript as Supplementary Datasets.

9. The authors use TEV-eGFP-tagged expression vectors throughout the paper. To exclude interference of the rather large eGFP tag, a C-terminus and N-terminus variant of the tag should be used and consideration should be given to repeat experiments with a 3xFLAG or V5-tagged vector.

The experiments were validated using FLAG-FAM46C, FAM46C-FLAG, GFP-FAM46C and NHA-FAM46C constructs (Extended Data Figure 4).

10. Statements on page 3, line 59 "a gene whose function is unknown"; page 3, line 74 and line 77-78 and 80-81 should be modified to acknowledge information provided by the above mentioned published material.

The manuscript has been modified accordingly.

11. Statement on page 4, line 90 is inaccurate. Please see Kuchta et al, and modify accordingly.

The word metazoans was changed into animals.

Reviewer #3 (Remarks to the Author): Expert in translation/cancer

In this current study, Mroczek et al. identify FAM46C, a gene encoding a novel poly(A) polymerase and investigate its binding partners, substrates and potential role in multiple myeloma. They show that FAM46C poly(A) polymerase activity is important for the stability of its targets and is dependent on its catalytic activity. They provide some evidence proposing that the enzymatic activity of this protein is important for the cell survival in different MM cell lines. Additionally, they also identify the RNA targets of FAM46C and validate some of them. Despite the fact that they could not provide any hints of a molecular mechanism accountable for the specificity of FAM46C to target a subset of mRNAs, they characterize a novel poly(A) polymerase and highlight its importance for cell survival. In conclusion, although the authors have original concepts, some of the experiments lack significant controls. Major revision should be in place before this manuscript can be considered for publication.

Major Concerns:

In Figure 1a, the authors perform an in vitro polyadenylation assay with purified WT and mutant FAM46C protein. They should also provide the gel showing equal expression of purified proteins, in order to conclude that the catalytic mutant is deficient in polyadenylation. As they discussed in the text, the activity of FAM46C is very low. Did the authors try to change the purification methods or try to purify the protein from cells by pull down and repeat the polyadenylation assay with the cell extract?

Fam46 proteins have a general tendency to aggregate and are very difficult to work with. We have made numerous attempts to purify FAM46 proteins from *E. coli* and from animal cells in culture.

They are using FAM46D as a control, however they do not give any information about this protein. Are they similar?

This is the best we could do. FAM46C and FAM46D are very similar as can be seen in the alignments that are shown in the Extended Data Figure 1.

Is this gene also mutated in MM? This will be important to address in the manuscript. FAM46D is not mutated in MM. It is simply slightly more soluble than FAM46C.

In Figure 2, the authors used the tethering assay, which is widely accepted as a powerful tool to investigate the role of a protein of interest in mRNA stability. However, the authors did not use proper controls for the assay. As R-Luc reporter is transfected, another reporter, for example F-Luc, should be used as an internal control, and the RNA and protein levels (generally measuring the luciferase activity) should be normalized accordingly.

Done

The internally transfected reporter should also be used for the control in Northern blots, instead of methylene blue staining.

Internal reporter was quantified by RT-qPCR

Additionally, an R-Luc reporter that does not contain any Box B element should also be used as negative control, as some LN tagged protein has intrinsic activity on this reporter.

We have performed a similar experiment using FAM46C devoid of λ N domains. Furthermore, an additional control, in which we have used the RL5boxHSL+HhR reporter with a cyclic phosphate at the 3' end generated by a hammerhead ribozyme. The data are now included in the modified Figure 3.

Moreover, the authors should also provide the quantification of at least 3 independent experiments for Northern blots and show the standard deviations. In Figure 2f, the half-life of the reporter should also be calculated.

Requested quantifications of both protein and RNA were performed and are included as parts of a Figure 3 and Extended Data Figure 2d

In Figure 3, the authors stably overexpress GFP tagged version of WT and mutant FAM46C. They should provide a western blot showing the relative expression of overexpressed protein compared to the endogenous one.

As described above we were unable to measure endogenous FAM46C protein in MM cells.

This is especially important for Raji and HL60 cells, to be sure that the proteins are actually expressing.

FAM46C is expressed in Raji and HL60 cells as visualized by confocal imaging and western blot (see the answer to the Reviewer 2).

In Figure 3a, according to the text, the authors overexpressed GFP alone as a control, however the cells do not express GFP (GFP+: 0.3%), or perhaps there is labeling problem. In the same figure, the intensity of GFP-WT protein is very low, so it is crucial to check the expression of the protein at different stages. It is possible to have weak expression because the cells are dying, however the expression should be checked in an early point. Otherwise, the experiment is not conclusive.

We think that the our original description of the experiments was confusing. Mostly because the word control was used twice with different meanings:

- In a panel a, control meant non transduced cells.
- In planes b and c, control meant cells transduced with GFP alone.

We have modified description to make the easier to follow. “Control” in panel 3a (4a in a revised version) has been change into non-transduced cells.

To verify the role of FAM46C as an onco-suppressor, as mentioned in the title, did the authors perform any knock-down assays of FAM46C to test whether this contributed to cellular transformation?

In order to provide additional evidence for FAM46C being an onco-suppressor we have performed shRNA mediated silencing and generated FAM46C KO animals. Please see the answer to the Reviewer 2

In Figure 5, the authors perform a pull down to identify interacting partners of FAM46C. Did they perform this IP in the presence of RNases? If not, how are they sure that the interaction is not mediated via RNA?

The IP was performed with RNAase treatment.

More importantly, how do they explain that PABP is unidentifiable in their MS, despite the high abundance of the protein.

We do not have a good explanation for it, but mass spectrometry is a methodology which has several biases and sometimes even if the protein is present in the sample may not be identified by this procedure.

If this is because of the IP condition, did they try to change the condition accordingly?

We have tried to increase the chance to identify weak/transient interactions by using DSP cross-linking procedure but unfortunately it did not reveal any interesting new hits.

Do they validate the interaction with the other proteins?

We have focused on the most promising hits based on the specificity and potential molecular functions.

Minor Concerns:

In the introduction, the authors state: “The entire sequence is altered by mutations except for the short 5’ coding.” What does this mean? Do they mean that the 5’ region of the CDS is not altered or that the 5’ UTR is not altered? They should describe this more clearly.

The sentence has been rephrased.

The authors should also summarize their results and the novelty of their findings properly in the introduction.

Done

In Figure 1d, the authors declare that FAM46C is expressed in the nucleus and cytoplasm and mention that the protein may have RNA stabilizing and destabilizing functions. However, they provide very poor quality confocal images. These images should be replaced, and control cells (without GFP) should also be included.

We have repeated the localization experiments (see above).

In Figure 2e, the control does not reflect the quantification.

Figure 2e (Extended Data Figure 2c,d in a new version of the manuscript) has been modified and GAPDH mRNA was used as a control for normalizations.

Lane 150: Is it MM1.S or RPMI8226?

Actually, RPMI8226 cells express wild-type FAM46C while MM1.S cells harbor mutation. Thus our statement is correct but since we have not performed any further experiments with MM1.S, it was removed from the manuscript to avoid confusions.

In Fig 5b, the asterisk should be explained.

Done

In the text, it would be easier for the reader to follow if the authors described the only mutant they use once, and then just label it as “mut”.

Done

Reviewers' comments:

Reviewer #1 (Remarks to the Author):

The authors should be congratulated on addressing several of the points raised in my review of the earlier version of their manuscript. In particular, they now include data from a newly derived FAM46C-FLAG knock-in mouse model, which add weight to the claim that the endogenous FAM46C protein is both nuclear and cytoplasmic, though the data presented in Fig. 5b are not of a very high quality (what is the evidence that the numerous bands marked with asterisks are non-specific?). The following outstanding points should nonetheless be addressed before publication:

1. The description of FAM46C as a tumor suppressor is rather premature, in my view, given the rather preliminary nature of the data. The latest version of COSMIC (v79, 14 November 2016) lists 104 unique samples with FAM46C mutations; of these, 30 are synonymous, 73 mis-sense and one nonsense. No frameshift mutations are reported. This is in stark contrast to the authors' statement in the first paragraph of the Introduction. The reason for this discrepancy should be clarified.
2. (a point raised in my earlier review that was not addressed by the authors in their rebuttal) In relation to the data shown in Fig. 4 (previously Fig. 3), what contributions might the numerous genetic differences between the cell lines tested (as opposed to the tumor type from which they were derived) make to their response to FAM46C expression?

Reviewer #2 (Remarks to the Author):

I have read with interest the revised version of the manuscript titled "The FAM46C gene encodes a non-canonical poly(A) polymerase and acts as an oncosuppressor in multiple myeloma" by Mroczek et al, currently under consideration for publication in Nature Communications.

The authors addressed the majority, if not all the points raised during the initial round of reviews. I would ask the authors to add a few details regarding their FAM46-FLAG knock out mice. Specifically, they should state how long mice were aged for and if any lymphoproliferative disorder or multiple myeloma like disorder developed at any given point.

My overall recommendation is to accept the paper for publication.

Reviewer #3 (Remarks to the Author):

The authors have addressed all of my concerns and the manuscript is suitable for publication.

Reviewer 4:

this reviewer commented for the editor only. He/she pointed out that you should provide the gel showing equal expression of purified proteins in Figure 1a in order to conclude that the catalytic mutant is deficient in polyadenylation.

In response to the reviewer 3 concern regarding the purification of FAM46 reviewer 4 suggest adding a gel showing expression levels of WT and FAM46C protein.

Finally, this reviewer suggested, given the similarity between FAM46D and FAM46C to see if FAM46D rescues the effects of FAM46C mutant expression.

Mroczek et al., point by point response to the Reviewer's comments

Reviewers' comments:

Reviewer #1 (Remarks to the Author):

The authors should be congratulated on addressing several of the points raised in my review of the earlier version of their manuscript.

We thank the reviewer for the positive opinion about our paper.

In particular, they now include data from a newly derived FAM46C-FLAG knock-in mouse model, which add weight to the claim that the endogenous FAM46C protein is both nuclear and cytoplasmic, though the data presented in Fig. 5b are not of a very high quality

We agree that the quality of our western blots is not outstanding. This is mainly because the protein we analyze is expressed at the endogenous level. We have screened several commercially available anti-FLAG antibodies and the result presented in Fig. 5b is the best what we could get.

(what is the evidence that the numerous bands marked with asterisks are non-specific?).

The bands marked by asterisks are non-specific since are present in samples isolated from both WT and FAM46C-FLAG knock-in animals. Such a statement is now included in the figure legend of the revised version of the manuscript.

The following outstanding points should nonetheless be addressed before publication:

1. The description of FAM46C as a tumor suppressor is rather premature, in my view, given the rather preliminary nature of the data. The latest version of COSMIC (v79, 14 November 2016) lists 104 unique samples with FAM46C mutations; of these, 30 are synonymous, 73 mis-sense and one nonsense. No frameshift mutations are reported. This is in stark contrast to the authors' statement in the first paragraph of the Introduction. The reason for this discrepancy should be clarified.

Our reference to COSMIC database was rather unfortunate since for the reason unknown to us the genomic data for multiple myeloma are rather underrepresented there. Based on the data published so far FAM46C is mutated in more than 10% of MM cases (<https://research.themmr.org>) and there are more than 70 different mutations identified in MM (Please see the diagram below which represented data obtained till 2015). Large fraction of alterations represents frameshift or nonsense mutations. FAM46C mutations are specific to MM and thus missense and synonymous mutations found in other

cancers may represent so called “mutational noise” We have modified the manuscript accordingly.

2. (a point raised in my earlier review that was not addressed by the authors in their rebuttal) In relation to the data shown in Fig. 4 (previously Fig. 3), what contributions might the numerous genetic differences between the cell lines tested (as opposed to the tumor type from which they were derived) make to their response to FAM46C expression?

We apologize for not directly addressing this point in the rebuttal letter and focusing on the technical difficulties in measuring the endogenous levels of FAM46C protein. The simple answer for this question is that we do not know exactly why MM is particularly FAM46C sensitive and in which way the genetic differences in various cancer types will contribute to FAM46C sensitivity or resistance. We assume that since MM cells produce massive amount of antibodies which are secreted through ER and FAM46C positively regulates expression of ER targeted mRNAs, FAM46C activity lead to ER overload.

We have added a commentary in the revised version of the manuscript.

Reviewer #2 (Remarks to the Author):

I have read with interest the revised version of the manuscript titled "The FAM46C gene encodes a non-canonical poly(A) polymerase and acts as an oncosuppressor in multiple myeloma" by Mroczek et al, currently under consideration for publication in Nature Communications.

The authors addressed the majority, if not all the points raised during the initial round of reviews. I would ask the authors to add a few details regarding their FAM46-FLAG knock out mice. Specifically, they should state how long mice were aged for and if any lymphoproliferative disorder or multiple myeloma like disorder developed at any given point.

The FAM46C KO mice have been obtained relatively recently and we were in the process of building the large colony. We have not seen any signs of spontaneous lymphoproliferative disorder in FAM46C KO animals aged for 12 months so far. Since the proper aging experiment has not been performed yet we would prefer not to discuss this issue in the revised version of the manuscript.

Information about the age of mice used for B lymphocyte isolation and hematologic parameters characterization is now included in the revised version of the manuscript.

My overall recommendation is to accept the paper for publication.

We thank the reviewer for the positive opinion about our paper

Reviewer #3 (Remarks to the Author):

The authors have addressed all of my concerns and the manuscript is suitable for publication.
We thank the reviewer for the positive opinion about our paper

Reviewer 4:

this reviewer commented for the editor only. He/she pointed out that you should provide the gel showing equal expression of purified proteins in Figure 1a in order to conclude that the catalytic mutant is deficient in polyadenylation.

In response to the reviewer 3 concern regarding the purification of FAM46 reviewer 4 suggest adding a gel showing expression levels of WT and FAM46C protein.

The gel showing recombinant FAM46C proteins (both WT and catalytic mutant) is now included in the revised version of the manuscript.

Finally, this reviewer suggested, given the similarity between FAM46D and FAM46C to see if FAM46D rescues the effects of FAM46C mutant expression.

We agree with the reviewer that the possible functional interchangeability of FAM46C and FAM46D is an interesting question. However, since our paper focuses on FAM46C we believe that such experiments are outside of the scope of this paper.

Reviewers' comments:

Reviewer #1 (Remarks to the Author):

The authors have addressed some, but not all, of the points raised in my previous review. In particular, the clarification of the mutation frequency of FAM46C in multiple myeloma is very welcome.

In relation to the other points:

1. "(what is the evidence that the numerous bands marked with asterisks are non-specific?).
The bands marked by asterisks are non-specific since are present in samples isolated from both WT and FAM46C-FLAG knock-in animals. Such a statement is now included in the figure legend of the revised version of the manuscript."

My point was that, as Figure 5b does not include samples from a wild-type mouse for comparison, the authors' claim (that the bands marked with asterisks in Figure 5b are non-specific) is not supported by the evidence presented. The bands marked with asterisks in the nuclear and cytoplasmic extracts (Figure 5b) seem to be different from those seen in whole-cell extracts (Figure 5a), so comparison between the two is not particularly helpful. In summary, nuclear and cytoplasmic extracts from wild-type mice should be included as controls in Figure 5b.

2. "(a point raised in my earlier review that was not addressed by the authors in their rebuttal) In relation to the data shown in Fig. 4 (previously Fig. 3), what contributions might the numerous genetic differences between the cell lines tested (as opposed to the tumor type from which they were derived) make to their response to FAM46C expression?"

We apologize for not directly addressing this point in the rebuttal letter and focusing of the technical difficulties in measuring the endogenous levels of FAM46C protein. The simple answer for this question is that we do not know exactly why MM is particularly FAM46C sensitive and in which way the genetic differences in various cancer types will contribute to FAM46C sensitivity or resistance. We assume that since MM cells produce massive amount of antibodies which are secreted though ER and FAM46C positively regulates expression of ER targeted mRNAs, FAM46C activity lead to ER overload.

We have added a commentary in the revised version of the manuscript."

I may not have explained my point sufficiently clearly in my earlier review. Figure 4 shows the effects of FAM46C expression in five cell lines. Like all tumor-derived cell lines, these will differ in probably millions of ways from each other (SNPs, mutations, copy number variation and so on). Three of the five cell lines respond to FAM46c expression in a manner that leads to decreased proliferation/increased death. This could be because these three cell lines are derived from MM; alternatively, it could be because these three cell lines (quite independently of the tumour type from which they are derived) share genetic features that make them sensitive to FAM46C expression. Can the authors be confident that the correlation with MM origin is anything other than coincidence? What is the Chi-squared test statistic? If they were to use 100 tumor-derived cell lines representative of multiple tumour types in this experiment, would only the MM-derived cell lines respond to FAM46C expression?

Reviewer #4 (Remarks to the Author):

I found that the authors responded to the reviewer's comments in a satisfactory manner.

Reviewers' comments:

Reviewer #1 (Remarks to the Author):

The authors have addressed some, but not all, of the points raised in my previous review. In particular, the clarification of the mutation frequency of FAM46C in multiple myeloma is very welcome.

Mutation frequency of FAM46C gene in multiple myelomas (MM) varies from one study to another and has been summarized in several reviews^{1, 2, 3, 4}. The very high frequency of mutations in FAM46C gene makes it important for MM progression and it was independently correlated to a poor prognosis^{5, 6}. An example of such an analysis is presented below (taken from¹).

Figure 3 | **Most frequent somatic mutations in patients with multiple myeloma.** Mutation frequencies were calculated by averaging the data from three whole-exome sequencing studies comprising a total of 733 patients^{6,7,9}. MM, multiple myeloma; WES, whole-exome sequencing.

A suitable clarification of the mutation frequency of FAM46C in MM has been included in the new version of the manuscript and the reference list was updated.

In relation to the other points:

1. “(what is the evidence that the numerous bands marked with asterisks are non-specific?). *The bands marked by asterisks are non-specific since are present in samples isolated from both WT and FAM46C-FLAG knock-in animals. Such a statement is now included in the figure legend of the revised version of the manuscript.*”

My point was that, as Figure 5b does not include samples from a wild-type mouse for

comparison, the authors' claim (that the bands marked with asterisks in Figure 5b are non-specific) is not supported by the evidence presented. The bands marked with asterisks in the nuclear and cytoplasmic extracts (Figure 5b) seem to be different from those seen in whole-cell extracts (Figure 5a), so comparison between the two is not particularly helpful. In summary, nuclear and cytoplasmic extracts from wild-type mice should be included as controls in Figure 5b.

We apologize for misunderstanding the reviewer's intentions. In the revised manuscript Figure 5b has been changed and extended. All suggested controls such as fractionation of wild type cells are now included. Moreover, the polyclonal antibody (PA1-984B) used in this study, albeit not perfect, the only one (we have tested several) that is able to detect FAM46C-FLAG at physiological expression levels. The presence of non-specific bands depends on the tissue from which protein extract was prepared.

2. "(a point raised in my earlier review that was not addressed by the authors in their rebuttal) In relation to the data shown in Fig. 4 (previously Fig. 3), what contributions might the numerous genetic differences between the cell lines tested (as opposed to the tumor type from which they were derived) make to their response to FAM46C expression?"

We apologize for not directly addressing this point in the rebuttal letter and focusing of the technical difficulties in measuring the endogenous levels of FAM46C protein. The simple answer for this question is that we do not know exactly why MM is particularly FAM46C sensitive and in which way the genetic differences in various cancer types will contribute to FAM46C sensitivity or resistance. We assume that since MM cells produce massive amount of antibodies which are secreted though ER and FAM46C positively regulates expression of ER targeted mRNAs, FAM46C activity lead to ER overload.

We have added a commentary in the revised version of the manuscript."

I may not have explained my point sufficiently clearly in my earlier review. Figure 4 shows the effects of FAM46C expression in five cell lines. Like all tumor-derived cell lines, these will differ in probably millions of ways from each other (SNPs, mutations, copy number variation and so on). Three of the five cell lines respond to FAM46c expression in a manner that leads to decreased proliferation/increased death. This could be because these three cell lines are derived from MM; alternatively, it could be because these three cell lines (quite independently of the tumour type from which they are derived) share genetic features that make them sensitive to FAM46C expression. Can the authors be confident that the correlation with MM origin is anything other than coincidence? What is the Chi-squared test statistic? If they were to use 100 tumor-derived cell lines representative of multiple tumour types in this experiment, would *only* the MM-derived cell lines respond to FAM46C expression?

We agree that MM cell lines, as well as primary MM tumors, are highly genetically heterogeneous^{7, 8, 9}. Thus, we do not have direct proof that FAM46C expression is toxic for all MM cells as we only analyzed selected cell lines. Moreover, we cannot exclude that FAM46C suppresses the growth of other cell types than ones derived from the B cell lineage. Actually, some recent studies from the Xin Lab suggest that FAM46C can also be involved in regulation of proliferation of hepatocellular carcinoma cells^{10, 11}. Importantly, however, FAM46C mutations are specific to MM (with incidence oscillating at about 10%) since statistically significant enrichment in FAM46C mutations has not been described for any other cancer type so far¹² (Lawrence MS et al. Nature 2014; see supplementary table 2 for details). Thus, we are quite confident that MM cells must benefit from mutations in FAM46C. The present paper confirms that indeed FAM46C is toxic for selected MM cell lines with FAM46C mutations since all analyzed MM cell lines were responsive to FAM46C expression. We also provide evidence that FAM46C is a general growth suppressor for the B cell lineage since primary B lymphocytes isolated from FAM46C^{KO} mice grow faster. However, FAM46C does not affect all cell types since the FAM46C^{KO} mice have no visible developmental defects. In conclusion, although the number of cell lines analyzed by us is not sufficient to use statistics to validate FAM46C toxicity for all MM cases, our data together with statistically significant enrichment of FAM46C mutations in MM allow us to claim that indeed FAM46C acts as a growth suppressor in MM, in agreement with earlier predictions based on high-throughput sequencing of MM cells¹³. Importantly, in this paper we focus on the function of FAM46C rather than simple evidence for its toxicity. The manuscript has been modified in order to clarify this issue.

Reviewer #4 (Remarks to the Author):

I found that the authors responded to the reviewer's comments in a satisfactory manner.

We thank the reviewer for the positive opinion about our paper.

References:

1. Manier S, Salem KZ, Park J, Landau DA, Getz G, Ghobrial IM. Genomic complexity of multiple myeloma and its clinical implications. *Nat Rev Clin Oncol* **14**, 100-113 (2017).
2. Lohr JG, et al. Widespread genetic heterogeneity in multiple myeloma: implications for targeted therapy. *Cancer Cell* **25**, 91-101 (2014).
3. Bolli N, et al. Heterogeneity of genomic evolution and mutational profiles in multiple myeloma. *Nat Commun* **5**, 2997 (2014).

4. Walker BA, *et al.* Mutational Spectrum, Copy Number Changes, and Outcome: Results of a Sequencing Study of Patients With Newly Diagnosed Myeloma. *J Clin Oncol* **33**, 3911-3920 (2015).
5. Boyd KD, *et al.* Mapping of chromosome 1p deletions in myeloma identifies FAM46C at 1p12 and CDKN2C at 1p32.3 as being genes in regions associated with adverse survival. *Clinical cancer research : an official journal of the American Association for Cancer Research* **17**, 7776-7784 (2011).
6. Chang H, Qi X, Jiang A, Xu W, Young T, Reece D. 1p21 deletions are strongly associated with 1q21 gains and are an independent adverse prognostic factor for the outcome of high-dose chemotherapy in patients with multiple myeloma. *Bone Marrow Transplant* **45**, 117-121 (2010).
7. Corre J, Munshi N, Avet-Loiseau H. Genetics of multiple myeloma: another heterogeneity level? *Blood* **125**, 1870-1876 (2015).
8. Prideaux SM, Conway O'Brien E, Chevassut TJ. The genetic architecture of multiple myeloma. *Adv Hematol* **2014**, 864058 (2014).
9. Morgan GJ, Walker BA, Davies FE. The genetic architecture of multiple myeloma. *Nat Rev Cancer* **12**, 335-348 (2012).
10. Zhang QY, Yue XQ, Jiang YP, Han T, Xin HL. FAM46C is critical for the anti-proliferation and pro-apoptotic effects of norcantharidin in hepatocellular carcinoma cells. *Scientific reports* **7**, 396 (2017).
11. Wan XY, Zhai XF, Jiang YP, Han T, Zhang QY, Xin HL. Antimetastatic effects of norcantharidin on hepatocellular carcinoma cells by up-regulating FAM46C expression. *Am J Transl Res* **9**, 155-166 (2017).
12. Lawrence MS, *et al.* Discovery and saturation analysis of cancer genes across 21 tumour types. *Nature* **505**, 495-501 (2014).
13. Chapman MA, *et al.* Initial genome sequencing and analysis of multiple myeloma. *Nature* **471**, 467-472 (2011).

REVIEWERS' COMMENTS:

Reviewer #1 (Remarks to the Author):

The authors have satisfactorily addressed the points raised in my review of the previous version of this manuscript. In fact, by stating "the clarification of the mutation frequency of FAM46C in multiple myeloma is very welcome", I was indicating that I was quite happy with that aspect of the previous version; nonetheless, the further clarification offered in the most recent draft is similarly welcome.

The manuscript will benefit from copy editing to improve clarity in a number of places, but this should not present any particular difficulty.

I'd like to take this opportunity to congratulate the authors on their presentation of an important, ground-breaking study.

Mroczek et al., point by point response to the Reviewer`s comments

Reviewers' comments:

Reviewer #1 (Remarks to the Author):

The authors have satisfactorily addressed the points raised in my review of the previous version of this manuscript. In fact, by stating "the clarification of the mutation frequency of FAM46C in multiple myeloma is very welcome", I was indicating that I was quite happy with that aspect of the previous version; nonetheless, the further clarification offered in the most recent draft is similarly welcome.

The manuscript will benefit from copy editing to improve clarity in a number of places, but this should not present any particular difficulty.

I'd like to take this opportunity to congratulate the authors on their presentation of an important, ground-breaking study.

We thank the Reviewer for an overall positive opinion about our paper. We have edited the manuscript to improve the clarity according to the Reviewer's request.